# Repeated vaccination with homologous influenza hemagglutinin broadens human antibody responses to unmatched flu viruses

Yixiang Deng[1†], Melbourne Tang[2†], Ted M Ross[3,4], Aaron G Schmidt[1,5], Arup K Chakraborty[1,2,6,7,8]*, Daniel Lingwood[1]*

[1]Ragon Institute of Mass General, MIT, and Harvard, Cambridge, United States; [2]Department of Physics, Massachusetts Institute of Technology, Cambridge, United States; [3]Center for Vaccines and Immunology, University of Georgia, Athens, United States; [4]Department of Infectious Diseases, University of Georgia, Athens, United States; [5]Department of Microbiology, Harvard Medical School, Boston, United States; [6]Department of Chemical Engineering, Massachusetts Institute of Technology, Cambridge, United States; [7]Department of Chemistry, Massachusetts Institute of Technology, Cambridge, United States; [8]Institute for Medical Engineering and Science, Massachusetts Institute of Technology, Cambridge, United States

*For correspondence:
arupc@mit.edu (AKC);
DLINGWOOD@mgh.harvard.edu (DL)

†These authors contributed equally to this work

**Abstract** The ongoing diversification of influenza virus necessitates annual vaccine updating. The vaccine antigen, the viral spike protein hemagglutinin (HA), tends to elicit strain-specific neutralizing activity, predicting that sequential immunization with the same HA strain will boost antibodies with narrow coverage. However, repeated vaccination with homologous SARS-CoV-2 vaccine eventually elicits neutralizing activity against highly unmatched variants, questioning this immunological premise. We evaluated a longitudinal influenza vaccine cohort, where each year the subjects received the same, novel H1N1 2009 pandemic vaccine strain. Repeated vaccination gradually enhanced receptor-blocking antibodies (HAI) to highly unmatched H1N1 strains within individuals with no initial memory recall against these historical viruses. An in silico model of affinity maturation in germinal centers (GCs) integrated with a model of differentiation and expansion of memory cells outside GCs during a recall response provides insight into the potential mechanisms underlying these results and shows how repeated exposure to the same immunogen can broaden the antibody response against diversified targets.

## Editor's evaluation

How to develop the vaccination method to induce broadly reactive neutralizing antibodies against variant viruses such as influenza is now a central issue in this field. In this regard, this valuable study provides outlines the mechanism by which repeated vaccination broadens the breadth of antibody responses against epitope unmatched virus strains. The authors' mathematical model is solid and incorporates varipous parameteres that regulate B cell activation and antibody response.

**eLife digest** Vaccines are the most effective way to prevent many infectious diseases and millions of deaths worldwide every year. They work by training the immune system to react to a protein – the antigen – that is specific to a pathogen. In response, the body produces specific antibodies, large proteins that mark the invaders for destruction. This builds a memory of this particular pathogen, enabling them to fight future infections better.

Some vaccines contain antigens, while others contain weakened or inactivated viruses or bacteria. Newer vaccines often contain a blueprint for producing antigens, such as DNA or RNA, instead of an antigen.

Each year, the flu vaccine is updated to match the main flu strains in circulation. This is because the vaccine's key component – the spike protein hemagglutinin (HA) – works best when it triggers antibodies that recognize and neutralize viruses with the same HA sequence.

A major obstacle to creating a universal flu vaccine is that influenza viruses constantly mutate, weakening the match between the vaccine and the virus. The problem is made worse because vaccines tend to produce antibodies that target the very parts of HA that change most frequently.

Repeated vaccination with the same flu shot (called vaccine boosting) was thought to strengthen the original immune response by increasing the number of antibodies targeting the same variable parts of HA. However, recent findings from studies on SARS-CoV-2 (the virus that causes COVID-19) suggest this is not always the case. Instead, repeated vaccination can both boost existing antibodies and generate new ones that target previously unrecognized regions of the antigen. This broader antibody response can help protect against variant viruses that share these newly recognized regions.

Deng, Tang et al. tested whether this antibody broadening also occurs with repeated influenza vaccination. The researchers evaluated data from a group of people who received the same flu vaccine for four consecutive years between 2013 to 2016, which included the new 2009 pandemic strain that many had never encountered before. The analyses showed that over time, their antibody responses became more diverse and capable of recognizing flu viruses spanning almost a century of evolution.

To understand how this happens, Deng, Tang et al. built a computational model to trace how a type of immune cell known as the B cells mature and diversify their antibody production after vaccination. The findings suggest that the broad antibody response during vaccination boosting is an inherent feature of the human immune system.

The next step and challenge will be to harness the natural ability to broaden antibody responses for designing vaccines that can protect against strains that may emerge in the future. This could involve fine-tuning HA or other vaccine proteins to better guide the immune system toward producing a broadly protective set of antibodies. Importantly, this principle of immune broadening could apply to any vaccine antigen – not just influenza.

## Introduction

Seasonal influenza vaccines are designed to elicit protective antibody responses against the viral strains predicted to dominate during an upcoming winter season (*Comber et al., 2023*; *Fiore et al., 2009*; *Jordan et al., 2023*; *Sandor et al., 2021*). The vaccine is typically trivalent or quadrivalent and aims to cover human-infecting influenza A viruses (IAV) and influenza B viruses (IBV) (*Demirden et al., 2022*; *Reed et al., 2012*; *Soema et al., 2015*). This has included an H1N1 vaccine strain for group 1 IAV, an H3N2 vaccine strain for group 2 IAV, and Yamagata and/or Victoria lineages for IBV. Elicitation of antibodies engaging the receptor-binding site (RBS) on the influenza spike protein hemagglutinin (HA) to block viral attachment is considered a major source of protection and is routinely measured by the hemagglutination (HA) inhibition (HAI) assay (*Cox, 2013*; *Krammer et al., 2020*; *Pedersen, 2014*; *Spackman and Sitaras, 2020*). Antibody Fc effector functions also provide orthogonal immuno-protective activities (*Boudreau and Alter, 2019*; *Boudreau et al., 2023*; *DiLillo et al., 2016*).

A concern of seasonal influenza vaccines is the lack of universality, where vaccine coverage can be lowered by antigenic drift of the virus, or even more worrying, antigen shift leading to the emergence of pandemic flu strains (*Bedi et al., 2023*; *Krammer et al., 2018*). These limitations are also underscored by the fact that individual HA molecules tend to elicit strain-specific antibody binding or neutralizing activity (*Altman et al., 2018*; *Angeletti and Yewdell, 2018*; *Bedi et al., 2023*;

*Sangesland and Lingwood, 2021*). Here, sequential immunization with homologous influenza HA antigens typically serves to boost strain-limited humoral output (*Henry et al., 2018*; *Krammer et al., 2018*; *Krammer and Palese, 2013*; *Sangesland and Lingwood, 2021*). Addressing these deficits has been a basis for rationally designed immune-focusing concepts tasked with re-orienting humoral immunity upon immune subdominant sites of conservation on influenza HA (*Altman et al., 2018*; *Angeletti and Yewdell, 2018*; *Caradonna and Schmidt, 2021*; *Krammer et al., 2018*; *Sangesland and Lingwood, 2021*; *Wei et al., 2020*). These efforts include structure-based reconfiguration and presentation of conserved HA moieties, and sequential immunization with strain variant antigens to further promote expansion of B cell memory against the invariant sites (*Amitai et al., 2020*; *Angeletti et al., 2019*; *Boyoglu-Barnum et al., 2021*; *Caradonna et al., 2022*; *Nachbagauer and Palese, 2018*; *Ray et al., 2024*; *Sangesland et al., 2019*; *Yassine et al., 2015*). A number of these 'universal' vaccine candidates are at various stages of clinical evaluation (*Andrews et al., 2023*; *Nachbagauer et al., 2021*; *Widge et al., 2023*).

Notably, however, recent human SARS-CoV-2 vaccine data warrants reconsideration of the basic premise that sequential immunization with homologous antigens elicits strong but strain-limited humoral immunity (*Garcia-Beltran et al., 2022*; *Muecksch et al., 2022*; *Schmidt et al., 2022*). Three sequential vaccinations with the homologous Wuhan-strain glycoprotein spike antigen elicit neutralizing antibody responses against highly unmatched Omicron variants. Broad neutralizing activity via engagement of the SARS-CoV-2 RBS was acquired after the third vaccination, consistent with a diversification of the repertoire of the antibodies elicited (*Garcia-Beltran et al., 2022*; *Muecksch et al., 2022*; *Schmidt et al., 2022*). Both antigen presentation dynamics and epitope masking activities within B cell germinal centers (GCs) appear to play key roles in the emergence of this broadened antibody response (*Yang et al., 2023*).

In the present study, we evaluated whether diversification of antibody binding/neutralization breadth via sequential immunization with homologous antigen is a general principle that characterizes human humoral immune responses. Accordingly, we evaluated an influenza vaccine cohort of individuals sampled longitudinally over 4 years (2013–2016) (*Nuñez et al., 2017*). HAI was measured before and after vaccination in each year, using a virus panel composed of diverse influenza A and B viruses spanning almost 100 years of evolution (*Nuñez et al., 2017*). Importantly, this vaccine cohort closely followed the 2009–2010 H1N1 pandemic and included 4 years of repeat exposure to 'non-imprinted'/pandemic A/California/7/2009 (pHA) as the sole H1N1 vaccine strain. Annual vaccination boosted HAI to vaccine-matched virus but also to highly divergent H1N1 viruses, despite the strong lack of relatedness. Importantly, this broadening did not occur via initial memory recall but rather intensified gradually over the 4-year vaccination period within individuals that were devoid of initial back-boosting against historical H1N1 viruses. To define a mechanistic framework for this effect, we adapted and extended a previous in silico model that accounts for B cell affinity maturation within GCs and associated memory B cell differentiation and expansion outside GCs (*Yang et al., 2023*). Using this approach, we describe mechanisms that may underlie the broadening of antibody coverage. We find that the broadening of the response is determined by the interplay between enhanced antigen presentation and epitope masking in GCs after booster shots, germline B cell affinities for different HA epitopes, and the level of conservation of these epitopes in the vaccinating strain with those on different historical variants. In these contexts, the capacity to eventually elicit broadly reactive antibody responses using a single influenza vaccine strain is discussed.

## Results
### The RBS patch of the 2009 H1N1 pandemic virus is strongly divergent from prior influenza strains

We began by applying a structure-based approach to define amino acid variation within the RBS (the epitope patch responsible for conferring HAI) amongst diverse IAV (H3N2 and H1N1) and IBV, spanning almost 100 years of evolution (*Figure 1*, *Figure 1—figure supplement 1*; *Appendix 1—table 1*). We assessed amino acid relatedness of the residues comprising the entire HA ectodomain (*Figure 1A, B*, *Figure 1—figure supplement 1*), and then the RBS patch, as defined by the structures of four human broadly neutralizing RBS-directed antibodies (bnAbs), each in co-complex with HA (*Schmidt et al., 2015*; *Figure 1C, D*; *Figure 1—figure supplement 1*). The paratopes of these bnAbs

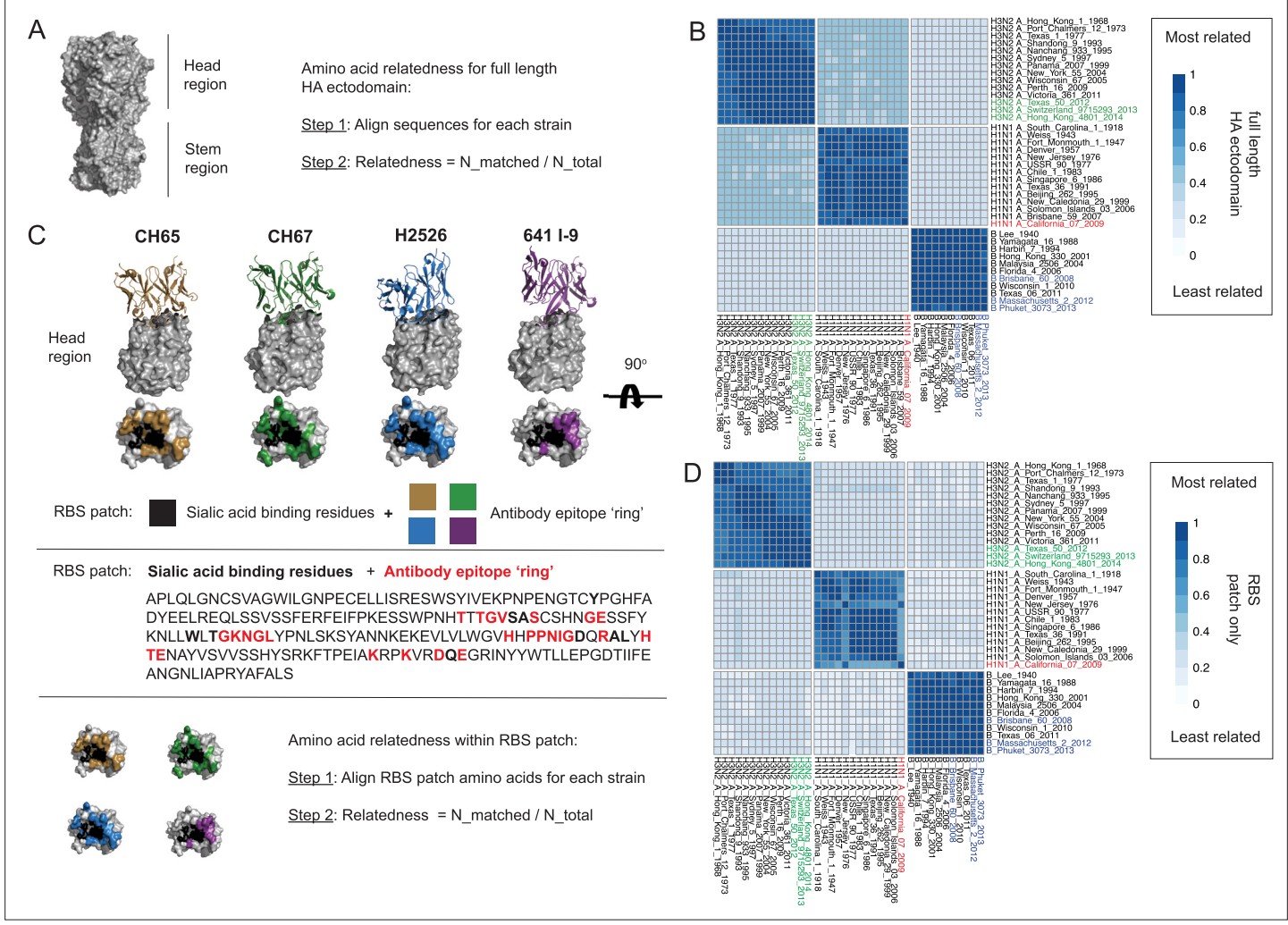

**Figure 1.** Divergent amino acid relatedness in the ectodomain and receptor-binding site (RBS) patch of the pandemic influenza HA. (**A**) The hemagglutinin (HA) ectodomain, where relatedness is calculated using the formula 'N_matched/N_total'; N_matched is the number of amino acids that match between the compared sequences and N_total is the total number of amino acids in the aligned sequence. (**B**) Heat map of HA ectodomain relatedness values for influenza A (H3N2, H1N1) and B viruses spanning almost 100 years (HA ectodomain sequences analyzed). (**C**) The RBS patch was structurally identified by four human bnAbs whose paratopes engage the RBS by mimicking cell surface sialic acid (CH67, CH67, H2526, 641 I-9) (**Schmidt et al., 2015**). We defined the RBS patch as the viral sialic acid binding residues (black) + the surrounding antibody epitope 'ring', collectively identified by the peripheral contacts made by the four bnAbs. Amino relatedness within the RBS patch is then calculated using the same formula except that the residues are now restricted to patch. (**D**) Heat map of HA RBS patch relatedness values for influenza A (H3N2, H1N1) and B viruses spanning almost 100 years (RBS patch sequences from the same 38 HA sequences as in **B**). See also **Figure 1—figure supplement 1** for extended resolution on the heat map scale.

The online version of this article includes the following figure supplement(s) for figure 1:

**Figure supplement 1.** Relatedness heat maps with extended resolution.

structurally mimic sialic acid, the primary receptor for influenza virus (**Schmidt et al., 2015**). In each case, the epitope footprint consists of the core viral amino acid residues responsible for binding sialyl oligosaccharide, along with a surrounding 'ring' of contact positions that are differentially engaged by the four bnAbs (**Figure 1C**, **Figure 1—figure supplement 1**). We defined the RBS patch as the sialic acid binding residues + the cumulative 'ring' of contact positions defined by these antibodies (**Figure 1C**, **Figure 1—figure supplement 1**). Amino acid relatedness values within the HA ectodomain and the RBS patch were then represented as heat maps for the influenza A and B viruses (**Figure 1B, D**, **Figure 1—figure supplement 1**). Within H1N1 viruses, the 2009 pandemic strain (pHA) stands out, along with A/New Jersey/1976, as strongly divergent, particularly within the RBS patch (**Figure 1B, D**).

This is consistent with previous reports on the structure of the RBS (*Cheung et al., 2020*; *Hong et al., 2013*; *Xu et al., 2010*) and the fact that both 2009 pandemic virus and the 1976 outbreak in Fort Dix, New Jersey, originated from swine lineages of H1N1 (*Garten et al., 2009*; *Gaydos et al., 2006*; *Mena et al., 2016*; *Sencer, 2011*; *Smith et al., 2009*; *Zimmer and Burke, 2009*).

## Sequential vaccination with homologous pHA broadly boosts HAI

To define how sequential immunization with homologous HA impacts antibody scanning breadth in humans, we evaluated the HAI coverage across the diverse viral strains from our relatedness analysis (*Figure 1B, D*), as elicited by repeated (4x) inoculation with influenza vaccine containing the same H1N1 component (pHA) over a 4-year period (2013–2016) (*Nuñez et al., 2017*; *Figure 2A*). Individuals were followed longitudinally (*n* = 136 individuals) (*Figure 2—source data 1*). In each year, a sample was obtained before and then 20 days after vaccination, and we first evaluated the fold change in HAI elicited against the virus panel (IBV, IAV H3N2, IAV H1N1) by each vaccine component within each year (*Figure 2B, C*). Notably, this analysis shows that pHA also elicits HAI for the highly unrelated/ historical H1N1 strains (i.e. full-length ectodomain relatedness <0.85; RBS patch relatedness <0.7).

## Sequential vaccination with homologous pHA broadens HAI with gradual kinetics in subjects that do not initially back-boost to historical strains

We next defined the kinetics of relatedness-independent broadening of H1N1 HAI over the 4-year period by graphing the fraction of responders versus non-responders (detectable vs non-detectable boosting of HAI to each H1N1 strain) at each year (*Figure 3A–C*). Although pandemic HA will not be historically imprinted, memory recall of pre-existing immunity or 'back-boosting' to historical strains would occur in response to the first antigen exposure (*Akkaya et al., 2020*; *Henry et al., 2018*; *Nuñez et al., 2017*; *Palm and Henry, 2019*; *Reusch and Angeletti, 2023*; *Turner et al., 2020*) and cannot be ruled out in the first vaccine year (2013). For this reason, we again focused on the initial non-responders, who boost against pHA (and the other seasonal vaccine components, see *Figure 2B, C*) but do not simultaneously broaden/back-boost against historical H1N1 strains post vax in Years 1 or 2, and by definition lacked B cell memory recalled by pHA (*Figure 3A–C*). The subsequent reduction of these non-responders upon sequential vaccination with pHA in later years identifies a separate vaccine broadening effect with slower kinetics (*Figure 3A–C*). In this effect, the proportion of non-responders to divergent H1N1 gradually decreases during the vaccine regimen, culminating in the near absence of non-responders in Year 4. This effect is seen when the subjects are not age stratified (*Figure 3A*) and when the subjects are divided into older and younger ages (>50 vs<38 years) (*Figure 3B, C*). The corresponding increases in the proportion of responders are also observed in these groups over the vaccine regimen (*Figure 3—figure supplement 1*). Collectively, these data indicate that within individuals that lack initial back-boosting, sequential exposure to pHA can eventually broaden the RBS-directed antibodies against highly unrelated H1N1.

## A computational model to study the mechanistic origin of increased coverage following sequential immunization with homologous HA

To obtain mechanistic insights underlying the observed broadening of the antibody response in the absence of prior immune imprinting and back-boosting, we adapted and extended a computational model of the humoral immune response upon repeated vaccination. The model is principally an extension of our past work modeling humoral immune responses upon repeated vaccination with SARS-CoV-2 vaccine immunogens, but also builds on our other past studies (*Amitai et al., 2020*; *Wang et al., 2015*; *Yang et al., 2023*). The purpose of this model is not to quantitatively fit clinical data, but to identify mechanistic principles that support the observations. Below, we outline the structure of the in silico model; mathematical and computational details are provided in the Methods.

We first coarse-grained the HA RBS into three antibody epitopes (epitopes 1–3) on pHA (strain 1) and on two historical H1N1 strains (strains 2 and 3) (*Figure 4A*). In this model, a fraction $p_i$ of the germline B cells target epitope *i*, and the immunodominance hierarchy is taken to be epitope 1 > epitope 2 > epitope 3. The immunodominant epitope on pHA (epitope 1) is taken to be very different (heavily mutated) in strain 1 as compared to strains 2 and 3. This is because the pandemic strain is expected to have changed substantially, so it would escape responses that target the immunodominant epitope

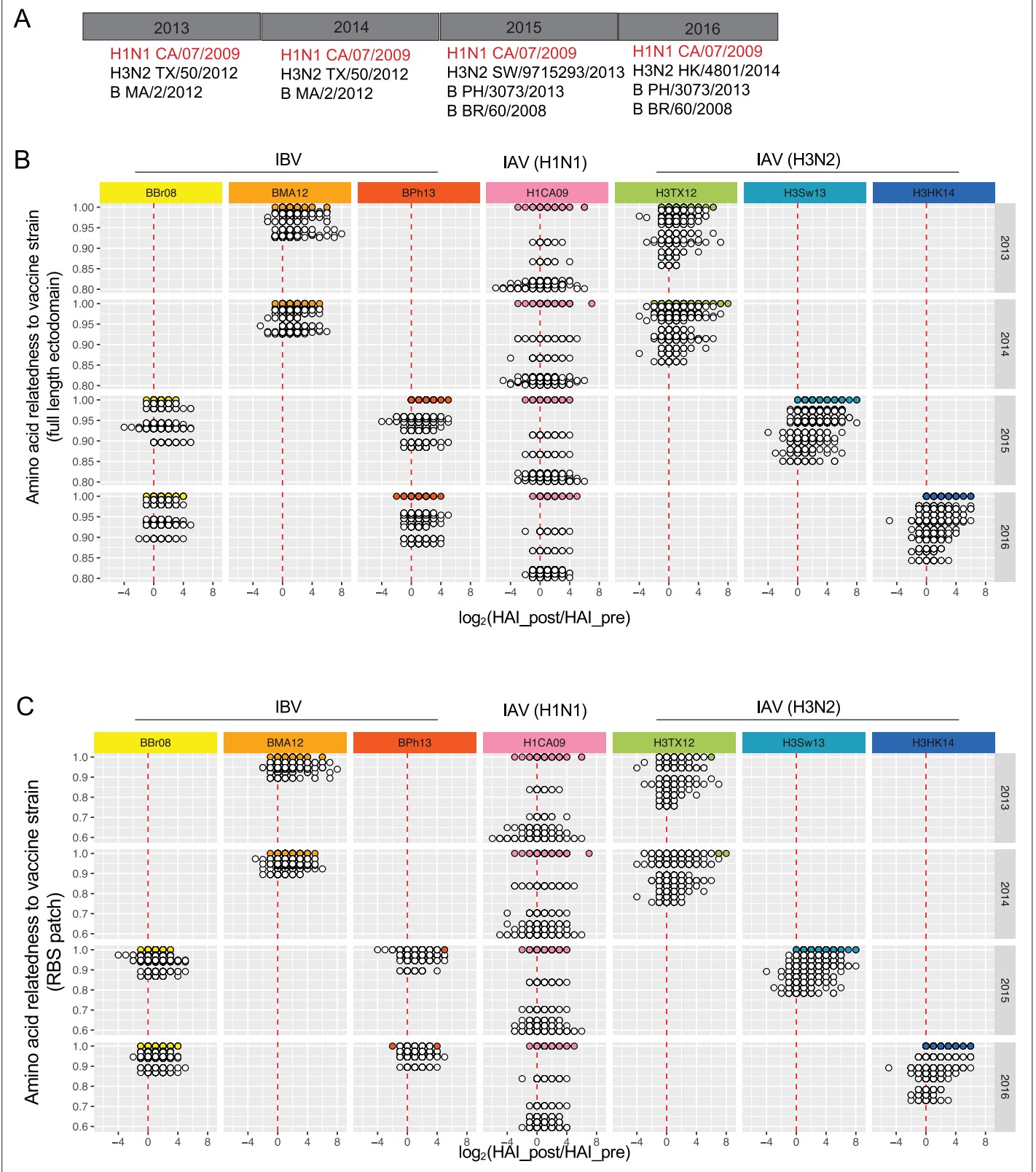

**Figure 2.** Sequential immunization with homologous pHA also elicits hemagglutination inhibition (HAI) against highly unrelated H1N1 strains. (**A**) Four-year influenza vaccine trial (**Nuñez et al., 2017**). We analyzed HAI elicited from subjects that were longitudinally followed and immunized each year with the vaccine strains indicated. Notably, these individuals received the same H1N1 component (A/California/07/2009 = pHA) in each of the 4 years. (**B**) Fold change in HAI titer (pre vs 20 days post-vaccination) elicited each year and graphed as a function of hemagglutinin (HA) ectodomain

*Figure 2 continued on next page*

*Figure 2 continued*

relatedness between the vaccine strain and the viruses within the HAI panels. Each dot is a single subject at the relatedness value: white dots are fold changes for strains from the virus panel; the colored dots indicate the vaccine-matched viral strain (relatedness = 1.00). (**C**) Same data as in (**B**) only now graphed as a function of receptor-binding site (RBS) patch relatedness between the vaccine strain and the viruses within the panels.

The online version of this article includes the following source data for figure 2:

**Source data 1.** Hemagglutination inhibition (HAI) values for the influenza virus strains, measured across longitudinal vaccine study (2013–2016) for *n* = 136 de-identified subjects >50 years of age and <38 years of age.

in historical strains. So, responses to a new immunodominant epitope need to evolve upon immunization with pHA. We further assume that subdominant epitope 2 is relatively conserved between strains 1 and 2, but not conserved between strains 1 and 3; subdominant epitope 3 is relatively conserved between strains 1 and 3, but not strains 1 and 2 (*Figure 4A*). In this way, we model three strains that are different from each other but share some similarities, as would be expected for all H1N1 viruses. Modeling only three H1N1 strains instead of dozens and just a few epitopes allows us to gain insights into the mechanism underlying the observed broadening of the response using a simpler computationally tractable system while still considering the complexities of generating immune responses to multiple strains.

The immunodominance hierarchy of the three epitopes is reflected in the distribution of germline B cell affinities for antigen, an attribute that is important for B cell recruitment into GCs (*Abbott and Crotty, 2020*; *Abbott et al., 2018*; *Amitai et al., 2020*; *Dosenovic et al., 2018*; *Sangesland and Lingwood, 2021*). Since high germline affinities are rare (*Feldman et al., 2021*; *Kuraoka et al., 2016*; *Lingwood et al., 2012*; *Ray et al., 2024*; *Ronsard et al., 2023*; *Sangesland et al., 2019*; *Sangesland et al., 2022*), we consider the distribution of affinities to be drawn from the tail of a distribution, which decays exponentially. The lowest affinity in the distribution we choose is $10^{-6}$ M, which is an estimate of the threshold for entry into GCs (*Batista and Neuberger, 1998*), which lies within the observed germline affinity range of human BCRs (*Feldman et al., 2021*; *Kuraoka et al., 2016*; *Lingwood et al., 2012*; *Ray et al., 2024*; *Ronsard et al., 2023*; *Sangesland et al., 2019*; *Sangesland et al., 2022*). More immunodominant epitopes constitute a larger fraction $p_i$ of germline B cells and exhibit a longer high-affinity tail (*Figure 4B*). We varied parameters that reflect the conservation of these epitopes and their relative immunodominance.

To then study humoral immune reactions to the different epitopes in silico, we modeled key steps that determine the antibody response pathway to protein antigens including: (1) (*Danecek et al., 2011*) antigen deposition on the surface of follicular dendritic cells (FDCs) (*Bhagchandani et al., 2024*; *de Silva and Klein, 2015*; *Victora and Nussenzweig, 2022*); (2) activation and entry of naive B cells into GCs, T helper-cell-driven affinity selection of B cells within GCs (*de Silva and Klein, 2015*; *Victora and Nussenzweig, 2022*; *Young and Brink, 2021*), and differentiation into memory B cells and plasma cells (*Akkaya et al., 2020*; *Crotty, 2015*; *Palm and Henry, 2019*); (3) relatively rapid expansion and differentiation of memory B cells into short-lived plasma cells during the recall responses which occur outside GCs (*Moran et al., 2018*; *Syeda et al., 2024*; *van Beek et al., 2022*); as elaborated in Methods, for brevity, we refer to these compartments outside GCs where memory cells are expanded in an antigen and T helper cell-dependent way as 'extra germinal centers' (EGCs).

A set of differential equations models the dynamics of antigen deposition and presentation on FDCs (see Methods for details). In the first few days after vaccination, soluble antigen rapidly decays in mice and non-human primates (NHP) (*Aung et al., 2023*; *Bhagchandani et al., 2024*; *Martin et al., 2021*). Circulating antibodies can bind to soluble antigen to form immune complexes (ICs) that become coated with complement, resulting in complement-receptor-dependent deposition of antigen on FDCs (*Phan et al., 2007*). ICs deposited on FDCs are longer lived than soluble antigen (*Aung et al., 2023*; *Martin et al., 2021*). For the first immunization, we assume that only weakly binding circulating antibodies are available for binding to the antigen and forming ICs. Therefore, relatively small amounts of ICs are deposited on FDCs before soluble antigen decays. For subsequent immunizations, stronger binding antigen-specific antibodies elicited by the previous immunization are available to bind antigen and form ICs before soluble antigen decays. This results in enhanced antigen deposition on FDCs. In vivo evidence for enhanced antigen deposition on FDCs mediated by antibodies generated at earlier time points has recently been provided by experiments in mice in the context of extended dosing of vaccine antigens (*Bhagchandani et al., 2024*). In our simulations,

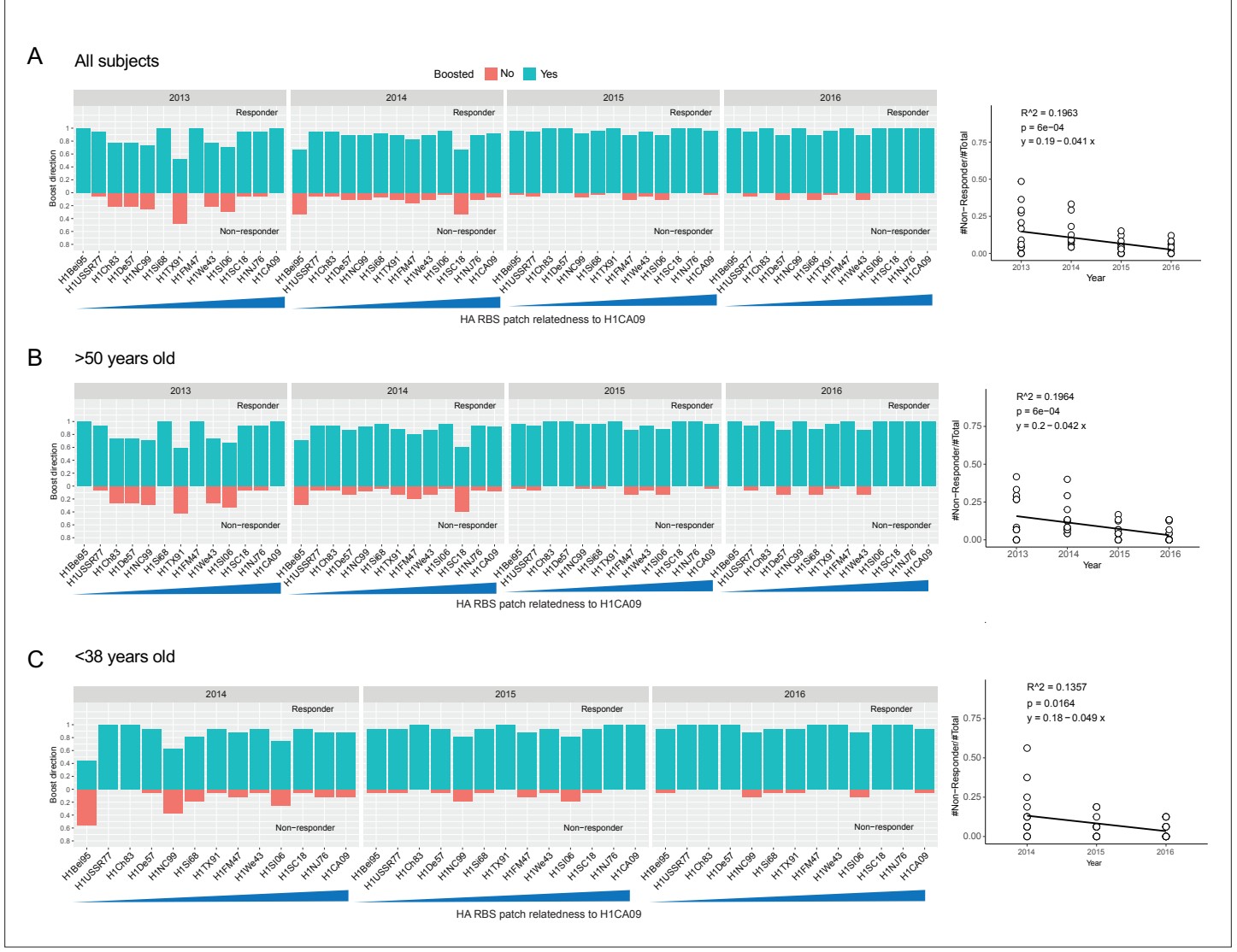

**Figure 3.** Sequential immunization with homologous pHA gradually broadens the response within individuals with no initial immune memory/recall to historical strains. Responders (green) versus non-responders (red) within each year is graphed for each H1N1 strain in the hemagglutination inhibition (HAI) panel. Responders are defined by having non-decreasing fold changes in HAI titers (post-vaccination HAI titer/pre-vaccination HAI titer; i.e. fold change >1). Non-responders are defined by having decreasing fold changes of HAI titers (post-vaccination HAI titer/pre-vaccination HAI titer; fold change <1). Because non-responders (red) do not back-boost against historical strains in the panel, they, by definition, lack imprinted immunity to these viruses that is recalled by pHA. In the regression analyses, each white dot denotes the proportion of non-responders for each viral strain. (**A**) Yearly response for all longitudinally analyzed individuals; at right is a linear regression of the proportion of non-responders over the 4-year vaccine data (p = 6e−04). (**B**) Data for subjects >50 years in age (p = 6e−04, linear regression). (**C**) Data for subjects <38 years in age (p = 0.0164, linear regression). See also *Figure 3—figure supplement 1* for linear regression of the proportion of responders in each age group.

The online version of this article includes the following figure supplement(s) for figure 3:

**Figure supplement 1.** H1N1 responders regressed over the vaccine.

the differential equations that describe these antigen and antibody dynamics are coupled to an agent-based simulation of the stochastic processes that occur inside GCs and when memory cells are expanded outside GCs during the recall response.

In the stochastic agent-based simulations of GCs and EGCs, each B cell is an agent and the probabilities of its activation by its B cell receptor's interactions with an epitope and antigen internalization, T cell-mediated selection, proliferation, mutation, and differentiation are calculated at each time step (0.01 days). Our model accounts for the following immunological principles and factors in the activation and selection of B cells: GC B cells internalize antigen based on their binding affinities

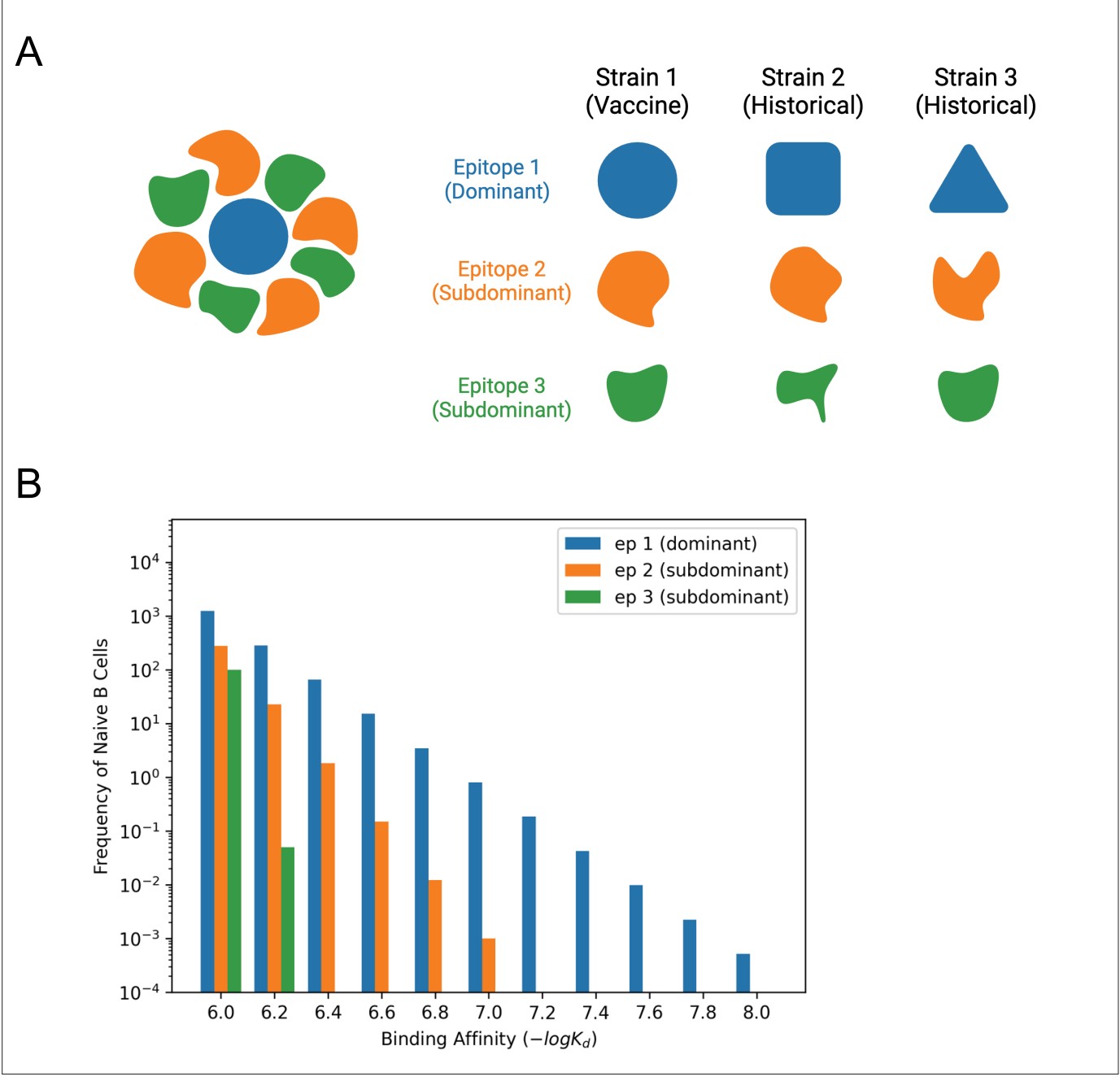

**Figure 4.** The influenza hemagglutinin (HA) head is coarse-grained into three epitopes that are perceived with different germline-endowed B cell affinities. (**A**) Diagram of epitope differences. In the right panel, the level of conservation of the three epitopes is depicted using different shapes (not very conserved) or similar shapes (relatively conserved). Epitope 1 (dominant epitope on pHA) is not conserved between the three variants. Epitope 2 (subdominant epitope) is relatively conserved between strain 1 (vaccine strain) and strain 2, but not between strains 1 and 3. Epitope 3 (another subdominant epitope) is conserved between strains 1 and 3, but not between strains 2 and 3. (**B**) Germline-endowed affinity distribution of naive B cells. Germline B cells targeting more dominant epitopes are more numerous and exhibit a longer high-affinity tail. Epitope 1 is more dominant than epitope 2, and epitope 2 is more dominant than epitope 3. Here, the fractions of naive B cells $p_i$ targeting epitope i are $p_1 = 0.8$, $p_2 = 0.15$, $p_3 = 0.05$.

to epitopes in the vaccine strain (*Batista and Neuberger, 1998*; *Fleire et al., 2006*); the amount of antigen internalized grows with the antigen binding free energy (or affinity) and saturates above a threshold affinity (*Foote and Eisen, 1995*; *Foote and Eisen, 2000*); and individual B cells that internalize antigen compete for subsequent T cell help to promote B cell survival (*Schwickert et al., 2011*; *Victora et al., 2010*). Among the B cells that are positively selected in the GC, some stochastically

exit the GC and differentiate into either plasma or memory B cells (*Akkaya et al., 2020*; *Crotty, 2015*; *Palm and Henry, 2019*). The majority of positively selected B cells are recycled for further mutation-selection cycles, and they proliferate and undergo somatic hypermutation (SHM) (*Collins and Jackson, 2018*; *de Silva and Klein, 2015*; *Glanville et al., 2009*; *Li et al., 2004*; *Mesin et al., 2016*; *Victora and Nussenzweig, 2012*; *Victora and Nussenzweig, 2022*). SHM is responsible for affinity-changing mutations, though it also leads to apoptosis or no affinity change with different probabilities (*Amitai et al., 2020*; *Wang et al., 2015*; *Yang et al., 2023*; *Zhang and Shakhnovich, 2010*). Based on data from experiments on affinity changes upon mutations at protein–protein interfaces, the change in affinity due to mutation is drawn from a log-normal distribution with only 5% of mutations being beneficial (*Kumar and Gromiha, 2006*; *Zhang and Shakhnovich, 2010*). Recent data in mice have also shown that ~5% mutations are beneficial (*DeWitt et al., 2025*). The mathematical details of the affinity-driven selection and SHM described above are summarized in the Methods.

In our model, memory cells are stochastically selected in an affinity-dependent way and expanded in EGCs via the same processes as in GCs, except that there are few to no mutations (none in our model) (*Moran et al., 2018*; *van Beek et al., 2022*; *Yang et al., 2023*). B cells exiting EGCs differentiate into antibody-secreting plasma cells with a probability of 0.6, because experimental data shows that 60% of new proliferating memory B cells differentiate into plasma cells (*Moran et al., 2018*). These plasma cells produce antibodies at rates estimated from clinical data on humans immunized with COVID vaccines (*Goel et al., 2021*; *Muecksch et al., 2022*). Our model also incorporates epitope masking in which circulating antibodies specific for a given epitope can enter ongoing GCs and EGCs and compete with B cells specific for the same epitope (*Bergström et al., 2017*; *McNamara et al., 2020*; *Schaefer-Babajew et al., 2023*; *Schiepers et al., 2024*; *Tas et al., 2022*; *Yang et al., 2023*; *Zarnitsyna et al., 2015*; *Zarnitsyna et al., 2016*).

To summarize, the key processes that occur in GCs in our model are mutations during SHM, together with differentiation of positively selected B cells into antibody-producing plasma cells and memory cells. In EGCs, memory cells expand in an affinity-dependent way during the recall response and produce many antibody-secreting plasma cells and more memory B cells.

While GC and EGC processes are driven by the vaccine antigen (strain 1), we also track the affinities of the resulting memory B cells and antibodies for strains 2 and 3 as well. A B cell's affinity for each strain depends on its initial affinity and the affinity-changing mutations that occur within the GC. The changes in affinity upon mutation are correlated between different strains. Specifically, the size of affinity changes is drawn from correlated log-normal distributions for the three epitopes, and the level of correlation is described by a parameter, $\rho$, that determines a covariance matrix. This parameter $\rho$ is related to the level of conservation and amino acid relatedness between the strains for the B cell's target epitope. For instance, if 70% of the amino acids in an epitope are shared between strains 1 and 2, we can approximate that ~70% of the beneficial mutations for B cells targeting this epitope in strain 1 are beneficial for strain 2 as well. The parameter $\rho$ for the corresponding epitope and strains is chosen to reflect the proportion of mutually beneficial mutations and hence quantifies the level of epitope conservation. We vary the values of $\rho$ to study the effects of different levels of epitope conservations.

For every immunization, we simulate 200 GCs and 1 EGC. The choice of 200 GCs was based on counts within spleen tissue section from immunized mice (~100 GCs per spleen) (*Jacob et al., 1991*) and would be predicted to be higher in the human secondary lymphoid organs. More details about this choice and the choice of one EGC per simulation are provided in the Methods. Ten different simulations are carried out for any given condition, and the results shown are averages over these simulations. The parameters used in the simulations are provided in *Appendix 1—table 2*.

## Booster shots of homologous pHA provide increasing coverage of historical strains through feedback regulation of the humoral response by modulation of antigen levels in GCs

*Figure 5A* shows the results of our simulations for antibody titers elicited against strain 1 (pHA) and the two historical strains (strains 2 and 3) after each of four immunizations with strain 1. The titers are calculated based on the affinity and number of antibodies that target the epitopes in each strain. After the first immunization, significant titers of antibodies are generated only against the dominant epitope of strain 1. However, after the second immunization, the titers are boosted against all strains, including

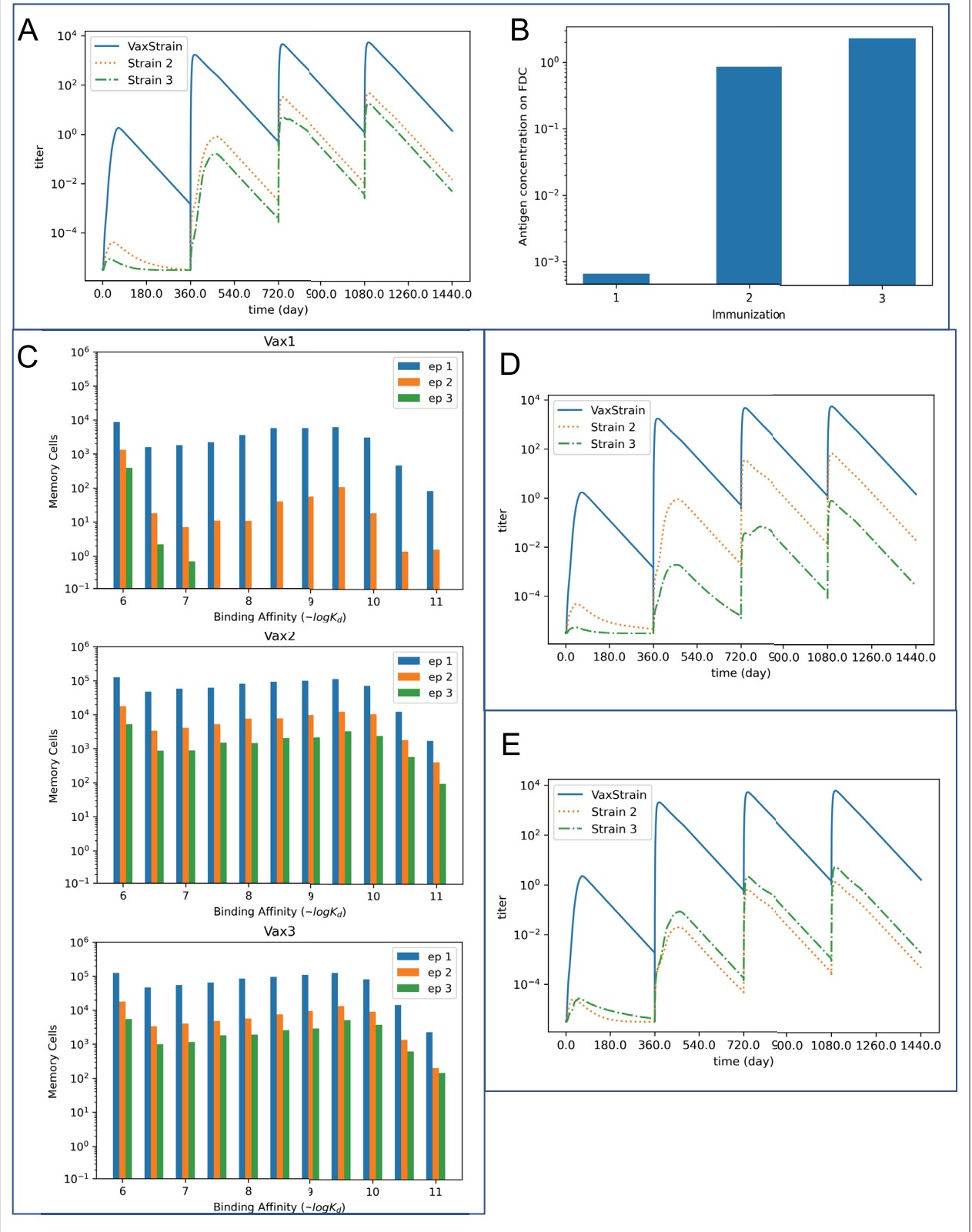

**Figure 5.** Antibody broadening via feedback regulation of the humoral response. (**A**) The antibody titers against both the vaccine strain and historical strains (strains 2 and 3) increase over four immunizations. The antibodies are produced by plasma cells from both the germinal centers (GCs) and the extra germinal centers (EGCs). Antibody coverage increases first for strain 2 (after the second immunization), and strain 3 is engaged after the third immunization. In this simulation, the initial fractions of B cells $p_i$ that target epitope i are $p_1 = 0.8$, $p_2 = 0.15$, $p_3 = 0.05$. The conservation $\rho_{12}$ of epitope

*Figure 5 continued on next page*

*Figure 5 continued*

2 between strains 1 and 2 and the conservation $\rho_{13}$ of epitope 3 between strains 1 and 3 are both equal to 0.95. (**B**) The expansion of pathogen-specific memory B cells from the first immunization and differentiation into plasma cells that produce antibodies significantly increases the antigen concentration on follicular dendritic cells (FDCs) in the second immunization. This allows lower-affinity B cells that target subdominant epitopes to enter GCs and undergo affinity maturation. The antigen concentration on FDCs slightly increases from the second to the third immunization, allowing more B cells that target the subdominant epitopes to enter GCs and undergo affinity maturation. (**C**) The distribution of memory cells produced in the GCs during the first three immunizations. Upon subsequent antigen exposure, these memory cells are selected and expanded in EGCs. Thus, they contribute significantly to circulating antibodies and increased titers during subsequent immunizations. The first immunization primarily produces memory cells that target the dominant epitope (epitope 1), along with some memory cells targeting epitope 2. The second and third vaccinations produce an overall greater number of memory cells bearing generally higher affinity for the subdominant epitopes (epitopes 2 and 3) than the first immunization. (**D**) Strain 3 is engaged less potently when the initial fractions of B cells $p_i$ that target epitope i are $p_1 = 0.8$, $p_2 = 0.18$, $p_3 = 0.02$. (**E**) The titers against strain 2 are lower than titers against strain 3 when the conservation of epitope 2 is decreased. Here the conservation $\rho_{12}$ of epitope 2 between strains 1 and 2 is 0.7 while the conservation $\rho_{13}$ of epitope 3 between strains 1 and 3 is kept at 0.95. Other values of $\rho_{12}$ are explored in *Figure 5—figure supplement 3*. The fractions of B cells $p_i$ that target epitope i are the same as those in **A**.

The online version of this article includes the following figure supplement(s) for figure 5:

**Figure supplement 1.** Booster shots of homologous pHA increase the number of high-affinity memory cells that target subdominant epitopes.

**Figure supplement 2.** Germinal centers (GCs) that form upon booster shots persist longer than those that form upon primary immunization.

**Figure supplement 3.** Weaker conservation of subdominant epitopes between historical strains and the immunizing strain can outweigh more favorable germline frequencies of B cells that target these epitopes.

**Figure supplement 4.** Increasing selection stringency K (from 0.5 to 0.7) does not change the qualitative results.

strains 2 and 3. Continued boosting with pHA continues to amplify heterologous coverage, even when the homologous boosting titer has plateaued. Experiments show that injected soluble antigen decays relatively rapidly (*Aung et al., 2023*; *Martin et al., 2021*; *Tam et al., 2016*). The vaccine antigen is the pandemic strain, and the individuals studied do not initially back-boost responses to historical strains. Therefore, upon the first immunization, only generic circulating antibodies, with low affinity for the antigen, are available to form the ICs needed for antigen deposition on FDCs (see computational methods section). Thus, our computational results show that very little antigen is deposited on FDCs after the first immunization with a new antigen (*Figure 5B*).

The germline B cells targeting the immunodominant epitope on strain 1 are more abundant and generally have higher affinities than the B cells engaging the subdominant epitopes. When antigen available on FDCs is low after the first immunization, the greater abundance and affinities of these germline B cells confer an especially strong advantage to them in entering GCs compared to B cells targeting subdominant epitopes. Furthermore, they are much more likely to dominate GC reactions during affinity maturation. Thus, the high-affinity memory B cells generated after the first immunization predominantly target the immunodominant epitope (*Figure 5C*, top panel; *Figure 5—figure supplement 1*).

After the second immunization, the EGCs facilitate the production of antibodies that engage strain 1, but not the historical strains. This is because the available memory B cells after the first immunization largely target the immunodominant epitope that is not conserved in the historical strains (*Figure 5C*, top panel; *Figure 5—figure supplement 1*). Thus, for many days after the second immunization, significant titers of antibodies that can target historical strains are not elicited (*Figure 5A*). However, secondary GCs also form during this time. Upon the second immunization, higher-affinity, antigen-specific IgG antibodies that were generated during the first immunization are available to bind and deposit the antigen on FDCs before the antigen is degraded for reasons noted earlier (*Bhagchandani et al., 2024*). This results in higher amounts of antigen deposited on FDCs (*Figure 5B*) and longer-lasting GCs (*Figure 5—figure supplement 2*). Higher amounts of deposited antigen allow lower-affinity germline B cells that target subdominant epitopes to enter the GC and be positively selected. As affinity maturation proceeds, these GC B cells acquire higher affinity to the subdominant epitopes that are relatively conserved between strain 1 and the historical strains, generating high affinity, subdominant-targeting memory B cells (*Figure 5C*, middle panel; *Figure 5—figure supplement 1*, middle and bottom panels). The plasma cells produced by the GCs produce antibodies that engage strains 2 and 3 (particularly strain 2 since epitope 2, the second most dominant epitope, is relatively conserved between strains 1 and 2).

After the third immunization, memory cells from the second immunization that target epitopes 2 and 3 with high affinity are also selected based on their affinities and expanded in the EGC. This results in further amplification of antibodies that target epitopes 2 and 3 (*Figure 5A*). Antigen deposition on FDCs is also somewhat elevated after the third immunization (*Figure 5B*). As a result, strain 3 is engaged more potently, and the relative number of GC-generated memory B cells targeting epitope 3 compared to epitope 2 increases after the third immunization (*Figure 5C*, bottom panel; *Figure 5—figure supplement 1*). Consequently, the difference between the titers produced against epitopes 2 and 3 is further decreased after the fourth immunization (*Figure 5A*). However, the overall improvements to antibody titers are minor because antigen presentation in the GCs and expansion of memory B cells in EGCs is similar to that after the third immunization.

The relative coverage of strains 2 and 3 upon repeated vaccination depends upon the relative immunodominance of epitopes 2 and 3 in the pool of germline B cells. To test the effects of modifying the immunodominance hierarchy, we increased the fraction of germline B cells that target epitope 2 and decreased the fraction that target epitope 3. This enhances the immunodominance of epitope 2 over epitope 3. Consequently, as shown in *Figure 5D*, the responses to epitope 3 and to historical strain 3 in which epitope 3 is conserved are less potent compared to the results shown in *Figure 5A*. Variation in the immunodominance patterns could explain variations in the kinetics of broadening against different historical strains as we observe in the clinical results (*Figure 3*, *Figure 3—figure supplement 1*).

The relative coverage of historical strains also depends on the conservation of subdominant epitopes between the historical and immunizing strains. In *Figure 5A*, the conservation of epitope 2 between strains 1 and 2 and of epitope 3 between strains 1 and 3 is the same. The results show that higher titers are elicited against strain 2 than strain 3, which aligns with the immunodominance of epitope 2 over epitope 3. However, it is possible that a more immunodominant epitope is less conserved as viruses mutate to avoid immune detection (*Altman et al., 2018*; *Angeletti et al., 2017*; *Angeletti and Yewdell, 2018*). Thus, we examined the effects of reducing the conservation of epitope 2 while fixing the conservation of epitope 3. If the conservation of epitope 2 between strains 1 and 2 is decreased below a critical value (see *Figure 5—figure supplement 3*), the advantage of epitope 2 due to a more favorable germline distribution is outweighed by weaker conservation. In *Figure 5E*, epitope 2 is more weakly conserved than in *Figure 5A* (but the germline immunodominance hierarchy is kept the same), resulting in lower titers against strain 2 than against strain 3. In this way, different levels of conservation of subdominant epitopes could explain why HAI activity against some strains is acquired before others during sequential immunization.

We also varied a parameter in our simulations that reflects the stringency with which GC B cells are selected in the GC based on their affinities for the antigen. We find that the qualitative results reported in *Figure 5* remain the same if this parameter is varied within a reasonable range (*Figure 5—figure supplement 4*).

## Feedback regulation of the humoral response by epitope masking enhances the generation of antibodies that can engage historical variants upon boosting with unmatched homologous HA

Circulating antibodies can enter GCs and bind to their corresponding epitopes on antigen presented on FDCs (*Bergström et al., 2017*; *Cyster and Wilson, 2024*; *McNamara et al., 2020*; *Schaefer-Babajew et al., 2023*; *Schiepers et al., 2024*; *Tas et al., 2022*; *Yang et al., 2023*; *Zarnitsyna et al., 2015*; *Zarnitsyna et al., 2016*). This masking of an epitope by soluble antibodies lowers the effective amount of antigen available to GC B cells targeting the same epitope, reducing their competitive fitness within the GC. This property of epitope masking by circulating antibodies can regulate the GC participation of naive B cells according to epitope specificity and modulate the competitive environment for GC B cells (*Bergström et al., 2017*; *Cyster and Wilson, 2024*; *McNamara et al., 2020*; *Schaefer-Babajew et al., 2023*; *Schiepers et al., 2024*; *Tas et al., 2022*; *Yang et al., 2023*; *Zarnitsyna et al., 2015*; *Zarnitsyna et al., 2016*). Our computational model shows that after the second immunization, most of the circulating antibodies bind to the dominant epitope. This is because most of the memory B cells produced after the first immunization are directed against the dominant epitope (*Figures 5C and 6C*). These memory cells are rapidly expanded in EGCs after the second immunization to generate the corresponding dominant epitope-targeting antibodies. Accordingly,

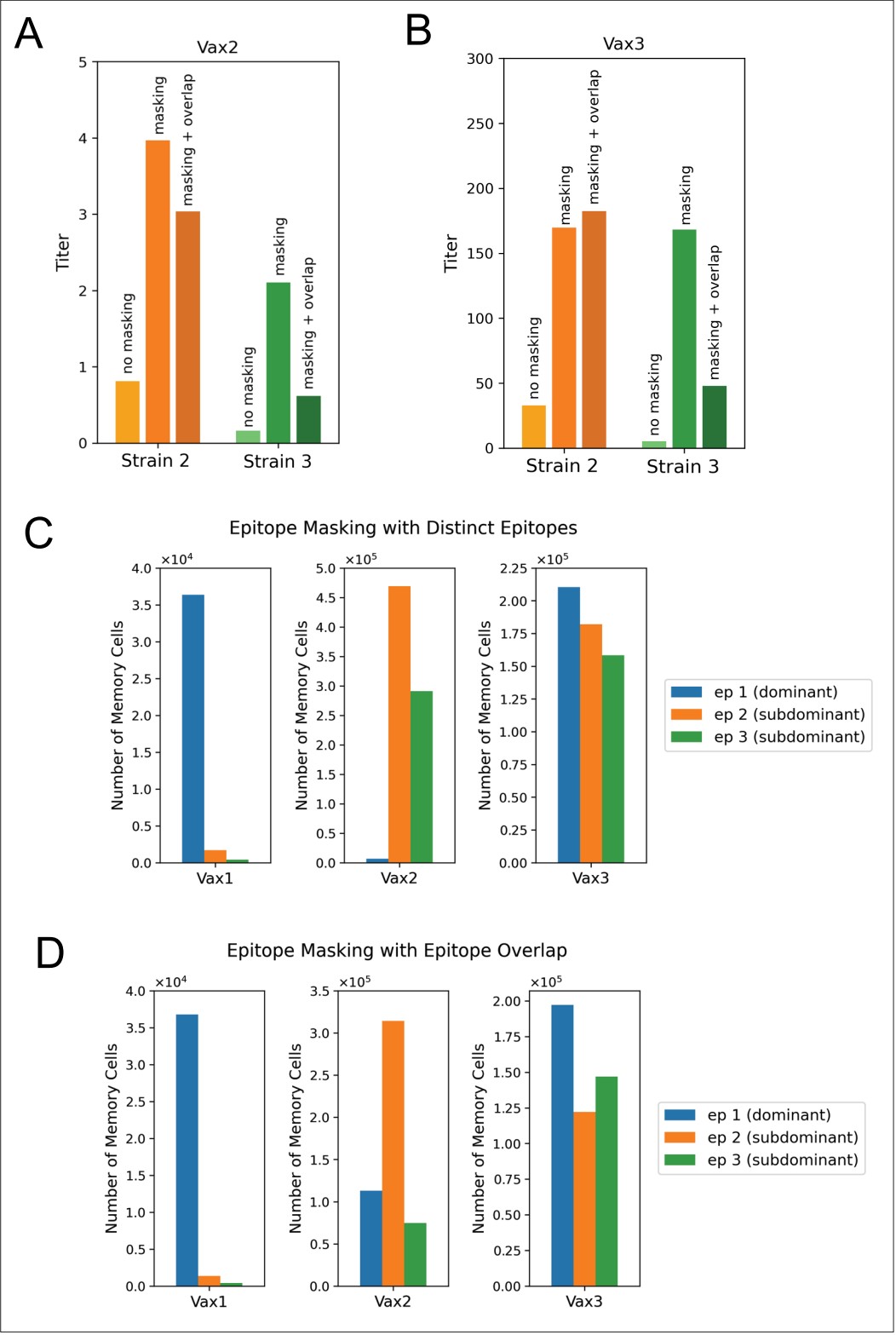

**Figure 6.** Regulation of antibody broadening through epitope masking. Maximum antibody titers for the historical strains after the second vaccination, with and without epitope masking. Two cases are considered when there is epitope masking: (1) (*Danecek et al., 2011*) the epitopes are absolutely distinct; (2) the epitopes can overlap with each other. In the second case shown here, there is 30% overlap between epitope 1 (dominant) and epitope 2 (subdominant) and between epitope 1 (dominant) and epitope 3 (subdominant). Masking increases the titers

*Figure 6 continued on next page*

*Figure 6 continued*

against historical strains, even when there is some overlap between the dominant and subdominant epitopes. (**B**) Maximum antibody titers for the historical strains after the third vaccination, with and without masking. After the third vaccination, titers for Variant 2 with epitope masking are higher when there is epitope overlap than when the epitopes are distinct. (**C**) Relative number of memory cells produced with epitope masking. Epitopes are considered to be fully distinct. The epitope that is most targeted by the memory cells is also masked the most after the subsequent immunization. The dominant epitope is targeted most by Vax 1 memory cells and is masked the most in the second immunization. The orange subdominant epitope (epitope 2) and green subdominant epitope (epitope 3) are both relatively well targeted by Vax 2 memory cells. However, the subdominant epitopes are also masked during the third immunization, so the subdominant epitopes lose their advantage compared to the dominant epitope in the affinity maturation process after Vax 3. (**D**) Relative number of memory cells produced with epitope masking and overlap. The epitope that is most targeted by the memory cells is masked the most in the subsequent immunization. The dominant epitope is targeted most by Vax 1 memory cells and is masked the most in the second immunization. The orange subdominant epitope (epitope 2) is targeted most by Vax 2 memory cells, although more memory cells target the dominant epitope than when the epitopes are fully distinct. Due to the masking of epitope 2 in the third immunization, the dominant and green subdominant epitope (epitope 3) are both relatively well targeted by Vax 3 memory cells.

we find that the entry and selection of subdominant-targeting B cells during affinity maturation are enhanced by the masking of the dominant epitope. As a result, more memory cells that target the subdominant epitopes are generated (***Figure 6C and D***) and larger antibody titers targeting strains 2 and 3 are elicited (***Figure 6A***) if epitope masking is included. Likewise, after the third immunization, the antibodies corresponding to memory cells from the second vaccination begin to mask epitope 2, promoting the generation of memory B cells and antibodies targeting epitope 3 (***Figure 6***). This feature of increasing antibody titers against the historical strains is observed when we consider two cases of epitope masking (***Figure 6A, B***): (1) (***Danecek et al., 2011***) where the three epitopes under consideration are non-overlapping; and (2) where the epitopes partially overlap with each other. The first case shows the largest effect of epitope masking. In the second case, epitope masking effects remain significant, even if we consider a modest level of epitope overlap (e.g. 30% between epitope 1 (dominant) and epitope 2 (subdominant); and 30% between epitope 1 (dominant) and epitope 3 (subdominant)).

Epitope overlap between dominant and subdominant epitopes generally decreases titers against historical strains compared to fully distinct epitopes. However, after the third vaccination, the antibody response titer against strain 2 is higher when there is epitope overlap (***Figure 6B***). This is because the second immunization produces more memory cells targeting the dominant epitope when there is epitope overlap (***Figure 6D***). These memory cells are rapidly expanded in EGCs, and so antibodies mask the dominant epitope more strongly during the third immunization when the epitopes overlap. This leads to the production of more high-affinity B cells that target epitope 2 after the third immunization, and subsequently higher titers against strain 2.

## Discussion

Our results reveal broadening of antibody coverage following repeated immunization with homologous pHA. This process occurred gradually within individuals that did not initially 'back-boost' against historical H1N1 strains. These findings are consistent with sequential exposure to SARS-CoV-2 mRNA vaccines, where repeated immunization with the homologous Wuhan vaccine strain eventually elicits non-imprinted coverage against highly unmatched Omicron lineages of SARS-CoV-2 (***Garcia-Beltran et al., 2022***; ***Muecksch et al., 2022***; ***Schmidt et al., 2022***). Our computational results suggest that feedback loops that regulate the recall humoral response (enhanced antigen presentation on FDCs and epitope masking) are a likely origin of eliciting non-imprinted antibody breadth to influenza HA. These mechanisms are consistent with the immune reactions that broaden antibodies after boosting with the homologous SARS-CoV-2 vaccine (***Yang et al., 2023***). More generally, our results point to a general feature of the humoral response that intrinsically broadens antibodies against unmatched/ diversified antigen targets upon repeated vaccination with the same novel antigen. The principle of preserving 'non-homogenized' antibody output with diverse epitope specificities and binding affinities has emerged as an important theme for GC and memory B cell responses to protein antigens (***de***

*Carvalho et al., 2023*; *DeWitt et al., 2025*; *Hägglöf et al., 2023*; *Kuraoka et al., 2016*; *Mesin et al., 2016*; *Mesin et al., 2020*; *Radmacher et al., 1998*; *Ronsard et al., 2023*; *Sabouri et al., 2014*; *Silver et al., 2018*; *Tas et al., 2016*; *van Beek et al., 2022*; *Zuo et al., 2025*) and our results suggest that intrinsic broadening of the antibodies generated in response to homologous antigen is an extension of this fundamental principle.

Antibody feedback is now a well-established mechanism that promotes or restricts GC recruitment depending on the concentration, affinity, and epitope specificity of the antibodies in circulation (*Bergström et al., 2017*; *Bhagchandani et al., 2024*; *Cyster and Wilson, 2024*; *McNamara et al., 2020*; *Schaefer-Babajew et al., 2023*; *Schiepers et al., 2024*; *Tas et al., 2022*; *Yang et al., 2023*; *Zarnitsyna et al., 2015*; *Zarnitsyna et al., 2016*). Our modeling suggests that such feedback regulation, which includes antigen presentation dynamics and epitope masking effects, can in part enable the polyspecific responses and broad B cell reactivities elicited following immunization with homologous immunogen. During the first exposure to antigen, generic circulating antibodies deposit limited amounts of antigen on FDCs as observed in experiments with NHP (*Martin et al., 2021*). Thus, mostly immunodominant responses result. Antigen-specific memory B cells generated in past exposures to antigen are expanded to generate antigen-specific antibodies that can deposit significantly more antigen on FDCs, consistent with studies on extended dosing vaccination in mice and NHP (*Bhagchandani et al., 2024*; *Cirelli et al., 2019*; *Tam et al., 2016*). Epitope-specific antibodies can also enter secondary GCs that form after re-exposure to antigen to mask their epitopes, thus increasing the relative concentration of other subdominant epitopes (*Bergström et al., 2017*; *Cyster and Wilson, 2024*; *McNamara et al., 2020*; *Schaefer-Babajew et al., 2023*; *Schiepers et al., 2024*; *Tas et al., 2022*; *Yang et al., 2023*; *Zarnitsyna et al., 2015*; *Zarnitsyna et al., 2016*). The overall enhancement in the amount of antigen driving GC reactions and epitope masking may promote the evolution of subdominant responses that target epitopes that are shared between the immunizing antigen and historical variants. Epitope masking will also be constrained by the dimensions of the RBS, and our simulations do report attenuation of titers against historical influenza strains when we introduce epitope overlap. Depending upon the degree of overlap between the epitopes and differences in germline characteristics in the B cells targeting dominant and subdominant epitopes, this effect could be compensated during subsequent shots, as seen in our results.

Operationally, it will be important to integrate this information to define the number of homologous immunizations needed to elicit a given level of vaccine coverage of historical or new strains. Our simulations indicate antibody broadening depends on the interplay of several key factors including: the precursor frequencies of germline B cells targeting these epitopes, the affinities of germline B cells for their target epitopes, and the level of conservation of shared epitopes within the RBS. Precursor frequencies and germline affinities determine the B cell pool's initial numbers and affinities toward target strains—historical or emergent—while conservation of shared epitopes governs how those affinities change during affinity maturation in response to selection of GC B cells for their affinities to the immunizing strain. Lower frequencies and affinities of germline B cells for conserved epitopes between vaccine and historical strains result in weaker antibody titers (compare strain 3 in *Figure 5A, D*) and poorer broadening for a fixed number of immunizations. On the other hand, increased conservation of shared epitopes can strengthen antibody titers (compare strain 2 in *Figure 5A, E*) and improve broadening for a given number of immunizations. Thus, the effects of the precursor frequencies and germline affinities can compete with the effects of epitope conservation in determining the coverage of a particular strain (*Figure 5—figure supplement 3*).

Our results also suggest that the germline B cell-mediated effects noted above are particularly important in earlier stages of the sequential vaccine regimen. But once higher affinity antibodies circulate (and recall of B cell memory becomes operational), epitope masking effects and enhanced antigen deposition on FDCs further modulate the immunodominance hierarchy to favor responses to subdominant epitopes. While our model does not account for memory B cells seeding secondary GCs, our previous work indicates that if more memory B cells enter secondary GCs, the evolution of broadened antibody responses is diluted (*Yang et al., 2023*).

In all cases, our simulations recapitulate the clinical vaccine results in a qualitative manner. This owes to our development of a simplified model for the vaccine antigen, its relation to historical strains, and the characteristics of the germline B cells that engage RBS epitopes. While recapitulation of antibody broadening through simplified computational models points to fundamental/core features of adaptive

immunity, more comprehensive epitope mapping of the B cell responses to the RBS targets (structure, engagement affinity, immunodominance patterns) will be needed to assemble more complete prediction maps to guide vaccine design.

In summary, we describe a phenomenon that defies the long-held view that sequential immunization with homologous influenza HA antigens serves to only/principally reinforce the boosting of antibodies with narrow coverage (*Henry et al., 2018*; *Krammer et al., 2018*; *Krammer and Palese, 2013*; *Sangesland and Lingwood, 2021*). Previous B cell epitope prediction algorithms have failed to computationally delineate and accurately predict patterns in the humoral response (*Mahanty et al., 2015*; *Rockberg and Uhlén, 2009*; *Sela-Culang et al., 2013*; *van Regenmortel, 2002*; *van Regenmortel, 2011*), but by modeling the key immunological steps that underscore adaptive immune reactions to protein antigen (*Akkaya et al., 2020*; *Crotty, 2015*; *de Silva and Klein, 2015*; *Palm and Henry, 2019*; *Victora and Nussenzweig, 2022*; *Young and Brink, 2021*), we obtain results in harmony with the clinical findings and describe mechanisms that may underlie them. This broadening effect has thus far only been observed in human antibody responses to vaccination, and we suggest that it may further inform universal vaccine approaches (*Caradonna and Schmidt, 2021*; *Krammer et al., 2018*; *Sangesland and Lingwood, 2021*; *Wei et al., 2020*). Immunologically, we suggest that antibody broadening reflects an in-built feature of continued B cell diversification, a principle that will ensure antibody complementarity to hypervariable antigen targets.

## Limitations of the study

Clinically, the vaccination regimen under study lacked an unvaccinated group, meaning that we cannot formally exclude the possibility that additional H1N1 infections also drove antibody broadening the 4-year vaccine period (2013–2016). However, pHA has dominated circulating H1N1 since 2009 (*Guthmiller and Wilson, 2018*; *Raymond et al., 2018*) and would be expected to contribute to broadening through the mechanisms we have described. We also are careful to report this broadening effect as an immunological feature of antigen scanning and do not make conclusions or assertions about its potential contribution to viral protection.

Computationally, model lays the groundwork to experimentally define factors identified as significant for antibody broadening in vivo, including epitope conservation, epitope masking, and antigen presentation on FDCs. We acknowledge that our computational model is one of many approaches that may be used to describe this effect and that it is not capable of quantitatively predicting outcomes of repeated vaccination because some parameters have not been defined with sufficient accuracy to allow building of general quantitative models of the humoral immune response. We also note that related models that we have employed in the past have been useful in obtaining mechanistic insights that have been experimentally tested in vivo (*Amitai et al., 2020*; *Bhagchandani et al., 2024*; *Wang et al., 2015*; *Yang et al., 2023*). We also mention that the number of EGCs (compartments where memory cells are expanded outside GCs) that form is not known. The use of a single EGC in each of our simulations may lead to greater stochastic variations between simulations that are used to calculate average properties. In the Methods, we note why such stochastic effects should not be pronounced in EGCs compared to GCs. The use of one EGC may also have effects on access to T cell help available if one used multiple EGCs, which may further modulate the pathways for broadening of the antibody responses that we describe.

While our model predicts that epitope conservation impacts titers against historical strains, the data examined in this study do not show a clear trend between amino acid similarity with the vaccine strain and the titer levels or coverage kinetics among H1N1 strains. Exploring alternate experimental metrics for epitope conservation instead of amino acid similarity, along with examination of diverse sets of clinical data, and measuring characteristics of germline B cells targeting different epitopes can improve the model and lead to better congruence between model predictions and data. In addition, experimentally defining the relative contribution of epitope masking and antigen presentation on FDCs will necessitate vaccine/immune challenge models where these parameters can be measured individually and that accurately reflect human humoral response features (*Aung et al., 2023*; *Schaefer-Babajew et al., 2023*; *Tas et al., 2022*; *Zarnitsyna et al., 2016*). In humans, an accurate description of individual imprinted B cell memory to influenza HA will also be needed to predict the broadening activity of homologous influenza vaccine regimen, in lieu of another pandemic influenza virus strain that does not back-boost to historical strains.

## Methods
### Lead contact and materials availability
Lead contact

Further information and requests for reagents should be directed to and will be fulfilled by Daniel Lingwood (dlingwood@mgh.harvard.edu) and Arup K. Chakraborty (arupc@mit.edu).

### Materials availability
There are no restrictions on the availability of the materials used in this study.

### Experimental model and subject details

We evaluated the HAI titers elicited by a clinical influenza vaccine that was sequentially immunized (4x) over a 4-year period (2013–2016) and contained the same H1 component (A/California/7/2009) in each of the vaccine years (*Nuñez et al., 2017*). An important distinction from the previous analysis is that we focused only on individuals that were longitudinally sampled across the 4-year period (*n* = 136 individuals; subjects are grouped into two age categories: above 50 years old and below 38 years old, see *Figure 2—source data 1*). In each year, a blood sample was obtained before and then 20 days after vaccination. HAI titers for viruses spanning 100 years of influenza evolution were measured in each of these samples (*Nuñez et al., 2017*; see *Figure 2—source data 1* and *Appendix 1—table 1*). The H1N1 viral panel for HAI comprised: A/South Carolina/1/1918, A/Weiss/JY2/1943, A/Fort Monmouth/1/1947, A/Denver/1/1957, A/New Jersey/6/1976, A/USSR/90/1977, A/Chile/1/1983, A/Singapore/6/1986, A/Texas/36/1991, A/Beijing/262/1995, A/New Caledonia/20/1999, A/Solomon Island/3/2006, A/Brisbane/59/2007, A/California/07/2009. The H3N2 vaccine panel for HAI comprised: A/Hong/Kong/1/1968, A/Hong/Kong/4801/2014, A/Nanchang/933/1995, A/New/York/55/2004, A/Panama/2007/1999, A/Perth/16/2009, A/Port/Chalmers/12/1973, A/Shandong/9/1993, A/Switzerland/9715293/2013A/Sydney/5/1997, A/Texas/1/1977, A/Texas/50/2012, A/Victoria/361/2011, A/Wisconsin/67/2005. The IBV viral panel for HAI included: B/Brisbane/60/2008, B/Florida/4/2006, B/Harbin/7/1994, B/Hong/Kong/330/2001, B/Lee/1940, B/Malaysia/2506/2004, B/Massachusetts/2/2012, B/Phuket/3073/2013, B/Texas/06/2011, B/Wisconsin/1/2010, B/Yamagata/16/1988.

### Method details
#### Amino acid relatedness in HA ectodomains or within the RBS patch

Amino acid sequences of HA ectodomains from the different strains used in HAI were obtained from Genbank (https://www.ncbi.nlm.nih.gov/genbank/) or GISAID (https://gisaid.org/; *Appendix 1—table 1*). The amino acid positions comprising the HA RBS patch were defined by the structures of four human broadly neutralizing RBS-directed antibodies, each in co-complex with HA (*Schmidt et al., 2015*). We used this information to define amino acid relatedness between the HA glycoproteins of all the influenza viruses used in our HAI panel. Pairwise relatedness was defined for both full-length HA ectodomain and for the RBS patch. In both cases, amino acid sequence relatedness was obtained by first aligning two amino acid sequences and then computing the ratio of matched amino acid counts over the total amino acid counts in the aligned sequences. Heat maps visualizing the pairwise amino acid sequence relatedness values were graphed using 'pheatmap' function from R package 'pheatmap' (version 1.0.12). Amino acid relatedness is represented as epitope conservation with the parameter $\rho$ in the computational model (see modeling section).

#### Analysis of HAI titers in relation to amino acid relatedness

We constructed dot plots to visualize the relationship between the fold change of HAI titers to the individual viruses in relation to their amino acid relatedness (for both the HA ectodomain and RBS patch) to the vaccine strain used in each year. In the case of the H1 vaccine component, the same H1 vaccine strain (A/California/7/2009) was used in each year. These plots were generated using the 'geom_point' function from R package 'ggplot2' (version 3.4.2). The fold change of HAI titer for each patient was computed by dividing the post-vaccination HAI titer by the corresponding pre-vaccination HAI titer.

## Longitudinal analysis of antibody broadening

To evaluate antibody broadening in response to homologous H1 (A/California/7/2009), we divided the HAI titers for H1N1 viruses from each individual subject into responders and non-responders to each viral strain in each year. Responders were defined by having non-decreasing fold changes of HAI titers (post-vaccination HAI titer/pre-vaccination HAI titer), that is fold change of HAI titers greater than 1. The non-responders were defined by having decreasing fold changes of HAI titers (post-vaccination HAI titer/pre-vaccination HAI titer), that is fold change of HAI titers less than 1. We tracked patients in the two age groups: >50 years old, having 4 years of complete HAI titers (2013–2016); and <38 years old, having 3 years of complete HAI titers (2014–2016). Bar plots showing the responder and non-responder ratios were graphed using 'geom_bar' function from R package 'ggplot2' (version 3.4.2). Linear regression analyses were also performed on the fraction of responders (or non-responders) in each year after standardizing the responder (or non-responder) value to the number of subjects in each age group (>50 years old or <38 years old). The linear regressions were performed using the 'ggscatter' function with the fitting equation shown by the 'stat_regline_equation' function from R package 'ggpubr' (version 0.6.0). Corresponding p and $r^2$ values were computed using the 'lm' function from R package 'stats' (version 4.3.1).

## Immune reactions in silico

The computational model is adapted from past work on the effects of repeated vaccination with COVID vaccines (*Yang et al., 2023*). Changes to the original model are described in the main text and below, along with key mathematical equations that describe the model. Interested readers can find in-depth rationale for model development, exploration of alternative model structures, and further analysis of parameter sensitivity in the earlier paper on the outcome of multiple immunizations with COVID vaccines (*Yang et al., 2023*). We use the same symbols to denote quantities as in the paper on COVID vaccines (*Yang et al., 2023*).

Differential equations describe antigen dynamics, and this is combined with stochastic simulations of GC and EGC processes. The time step is 0.01 day. For each situation, 200 GCs are simulated, and from the second immunization on, 1 EGC is simulated along with the GCs. Ten such simulations are carried out for each set of conditions, and the results are averaged over the ten simulations to report results.

The choice of 200 GCs was based on a paper that dissected sections of spleen in a 6 μm section and found that there were 100 GCs (*Jacob et al., 1991*). That is, there are an order of $10^2$ GCs. As the number 100 was for a part of mouse spleen and the number in humans is also likely higher, we used 200 GCs. The number of compartments outside GCs where memory B cells are expanded is unknown. Neither is it known how long they last after exposure to a shot of a vaccine. Using more EGCs, rather than one, would potentially reduce the effect of stochastic variations between simulations when we calculate averages over many simulations. Nevertheless, we expect stochastic effects to be less pronounced in EGCs compared to GCs, especially for the second shot. This is because the memory pool after the first shot is likely to be comprised of a few expanded clones and a representative sample of this pool is sampled for entry into the EGC. In contrast, the naive pool of germline B cells that can enter GCs is much larger, and this pool can be sampled better by modeling many GCs. In addition, stochastic effects within EGCs should also be less than in GCs because there is little to no mutation in the EGCs, while a usually rare mutation of relatively higher affinity could evolve in the GCs. However, in shots after the second one, stochastic fluctuations in EGCs cannot be ruled out. For providing qualitative mechanistic insights, using one EGC and averaging over many simulations should be reasonably adequate for calculating average properties.

### Antigen dynamics

Differential equations describe the reactions that govern the concentration of antigen and antibody, as shown in the table below. We use the following abbreviations and symbols: soluble antigen (Ag), soluble antibody (Ig), soluble immune complex (IC), immune complex on follicular dendritic cell (IC-FDC), plasma cell (PC), rate of decay (d), rate of reaction (k), dissociation constant of serum antibodies ($K_d$), dissociation constant of plasma cell antibodies ($K_d^{PC}$).

| Equation | Reaction(s) | Description |
|---|---|---|
| $\frac{[Ag][Ig]}{[IC]} = K_d$ | $Ag + Ig \leftrightarrow IC$ | Fast equilibrium for formation of immune complex |
| $\partial_t [Ag] = -d_{Ag}[Ag]$ | $Ag \rightarrow \varnothing$ | Decay of free soluble antigen |
| $\partial_t [IC] = -k_{deposit}[IC]$ | $IC \rightarrow IC - FDC$ | Immune complex transport to follicular dendritic cells |
| $\partial_t [IC - FDC] = k_{deposit}[IC]$ $\phantom{\partial_t [IC - FDC]} - d_{IC}[IC - FDC]$ | $IC - FDC \rightarrow \varnothing$ | Deposition and decay of antigen on follicular dendritic cells |
| $\partial_t [Ig] = k_{Ig}[PC] - d_{Ig}[Ig]$ | $PC \rightarrow PC + Ig$ $Ig \rightarrow \varnothing$ | Antibody production by plasma cells Decay of free soluble antigen |
| $\partial_t K_a = \frac{\left(K_a^{PC} - K_a\right)k_{Ig}[PC]}{[Ig]+[IC]}$ | - | Derived from equations above, as detailed in supplement of *Yang et al., 2023*. |

For the deposition of antigen on FDCs, it is assumed that the antibodies that bind to and deposit antigen have the same average antigen affinity and numbers as those whose evolution we explicitly simulate. The parameters used in these equations can be found in *Appendix 1—table 2* and are identical to our previous publication (*Yang et al., 2023*). Upon the first vaccination, only weakly binding antibodies are available for binding to soluble antigen and depositing ICs onto FDCs. After subsequent vaccinations, antigen-specific antibodies are available.

## B cell dynamics in GCs and EGCs

Each GC is associated with a pool of 2000 naive B cells (*Yang et al., 2023*). This estimate was based on estimating the number of germline B cells that target SARS-CoV-2, and we used the same number. Ideally, we would use values for HA if these measurements were available. A fraction $p_i$ of these naive B cells targets epitope $i$. We model 3 epitopes of the influenza spike protein, which is an increase from the 2 epitopes previously used to model the SARS-CoV-2 spike protein (*Yang et al., 2023*). This allows us to better account for the diversity in influenza strains, each with distinct epitopes that may be conserved with the H1N1 CA09 vaccine strain (see main text).

The germline binding affinities $E = -\log(K_d)$ of the naive B cells for the vaccine strain have discrete values between 6 and 8, expressed as $E_k = 6 + 0.287\,k$ for $k = 0, 1, \ldots, 10$. The lower value of 6 was chosen based on data as described in the main text, and germline affinities that are one hundred times greater have also been observed (*Yang et al., 2023*). The frequency of naive B cells targeting epitope $i$ in the affinity bin $E_k$ is a truncated geometric distribution and is determined as follows:

$$f_i(E_k) = N_{naive}p_i \frac{e^{-r_i(E_k - E_0)}}{\sum_k e^{-r_i(E_k - E_0)}},$$

where the number of naive B cells $N_{naive}$ is 2000 and the minimum germline affinity $E_0$ is 6 in these simulations. $r_i$ is determined such that

$$f_i(E_i^*) = p_i,$$

where $E_i^*$ is defined as follows using the parameters $E_1^h$, $dE_{12}$, and $dE_{13}$:

$$E_1^* = E_1^h,$$
$$E_2^* = E_1^h - dE_{12},$$
$$E_3^* = E_1^h - dE_{13}.$$

The parameters are set such that $E_1^* > E_2^* > E_3^*$, reflecting the generally higher affinities for more dominant epitopes. The immunodominance hierarchy is further modeled by setting, $p_1 > p_2 > p_3$ meaning that more dominant epitopes are targeted by a greater number of naive B cells.

Since individuals initially have a weak response (subdominant) against historical influenza strains, the germline-binding affinity of all naive B cells that target the epitopes that are conserved between the vaccinating strain and historical strains is set to the lowest possible germline affinity in these simulations ($E = 6$).

After initializing the pool of naive B cells, the B cells can be stochastically activated. The probability of activation for a naive B cell depends on the quantity of vaccine antigen it captures as it is influenced by the binding affinity of its BCR for antigen, which is determined by both the antigen concentration and binding affinity to the vaccine strain. The amount of antigen captured by B cell $j$ is modeled as

$$A_j = \left( \frac{C}{C_0} 10^{\min(E_j, 10) - E_0} \right)^K ,$$

where $C$ is the effective antigen concentration, $C_0$ is the reference antigen concentration, $E_j$ is the B cell's binding affinity for its epitope, $E_0$ is the reference affinity, and $K$ is a measure of the selection stringency. The effective antigen concentration is $C = 0.01 \left( [Ag] + [IC] \right) + [IC - FDC]$, reflecting that antigens presented on FDCs are more potent at activating B cells (**Kim et al., 2006**). The selection stringency $K$ represents how sensitive the amount of antigen captured is to small differences in antigen concentration or binding affinity. The probability of activation is $P \left( B\ cell\ j\ is\ activated \right) = \min(A_j, 1)$. The approach described above is used to calculate $P \left( B\ cell\ j\ is\ activated \right) = \min(A_j, 1)$ both for GC entry and inside the GC.

For GC entry, activated B cells can stochastically enter the GC. Entry into the GC depends on antigen captured and competition for limited T cell help (**Lee et al., 2021**; **Schwickert et al., 2011**). The rate of entry for an activated B cell $j$ is

$$\lambda_j = \frac{\dfrac{N_{max}}{N_{activated}} \dfrac{A_j}{\langle A \rangle}}{1 + \dfrac{N_{max}}{N_{activated}} \dfrac{A_j}{\langle A \rangle}} ,$$

where $N_{activated}$ is the number of activated B cells, $N_{max}$ is the capacity for GC entry based on limited T cell help, and $\langle A \rangle$ is the average amount of antigen captured by all B cells. Thus, $\frac{N_{max}}{N_{activated}}$ represents the competition between B cells for T cell help and $\frac{A_j}{\langle A \rangle}$ represents the competitive advantage of a particular B cell $j$ over other cells. The probability of GC entry is $P \left( B\ cell\ j\ enters\ GC \right) = 1 - e^{-\lambda_j dt}$.

After internalizing antigen and displaying pMHC molecules on the surface, GC B cells compete for T cell help to become stochastically activated. The rate of positive selection is

$$\beta_j = \beta_{max} \frac{\dfrac{N_T}{N_{activated}} \dfrac{A_j}{\langle A \rangle}}{1 + \dfrac{N_T}{N_{activated}} \dfrac{A_j}{\langle A \rangle}} ,$$

where $\beta_{\max}$ is the maximum rate of positive selection, $N_{activated}$ is the number of activated B cells, and $N_T$ is the number of helper T cells. We model the number of helper T cells $N_T$ as a simple linear growth until time $t_0$, when the number of helper T cells has a peak level of $N_{T0}$, and as first-order decay with decay rate $d_T$ after $t_0$:

$$N_T = \begin{cases} N_{T0}\ t/t_0, & t < t_0 \\ N_{T0}\ e^{-d_T(t - t_0)}, & t > t_0 \end{cases} .$$

If a B cell is positively selected, it exits the GC with probability $p_{exit}$ or is recycled for mutation-selection cycles in the GC with probability $1 - p_{exit}$. If the B cell exits, it becomes a plasma cell with probability $p_{plasma}$ or a memory cell with probability $1 - p_{plasma}$. If the B cell proliferates, one of the daughter cells mutates. The mutation may change affinity (probability 0.2), result in apoptosis (probability 0.3), or be silent (**Zhang and Shakhnovich, 2010**).

Each B cell has a string of 0 and 1 s for the residues on the paratope with a total length of $n_{res}$. The string of residues starts as all 0 s in a naive B cell. When there is an affinity-changing mutation, one of the bits (residues) is randomly chosen and flipped. The change in affinity is drawn from a shifted

log-normal distribution, independently for each residue (**Kumar and Gromiha, 2006**; **Zhang and Shakhnovich, 2010**). The affinity (binding free energy) of a B cell $j$ targeting a particular epitope, which may or may not be conserved in different strains, is determined by both the germline affinity and affinity-changing mutations, as follows ($E_j^{strain}$ refers to the affinity of B cell $j$ targeting a particular epitope on strain, $j$):

$$E_j^{strain} = E_j^{0,strain} + \sum_{k=0}^{n_{res}} \delta_{j,k} s_{j,k}^{strain}$$

where $E_j^{0,strain}$ is the germline affinity, $\delta_{j,k} \in \{0,1\}$ is the mutational state of residue $k$, and $s_{j,k}^{strain}$ is the change in affinity due to a mutation at residue $k$. $s_{j,k}^{strain}$ is correlated between different strains to different extents depending upon the degree of conservation of the epitope under consideration in these strains ($s_{j,k}^1$ is the change in affinity for the epitope in strain 1, $s_{j,k}^2$ is the change in affinity for the epitope in strain 2, and so on). The values of $s_{j,k}^{strain}$ are drawn from identical log-normal distributions that are correlated as follows:

$$\left[ s_{j,k}^1, s_{j,k}^2, s_{j,k}^3 \right] \sim - (\log_{10} e) \left( e^{N\left(\mu, \sigma^2 \Sigma\right)} - \epsilon \right)$$

$$\Sigma = \begin{bmatrix} 1 & \rho_{12} & \rho_{13} \\ \rho_{12} & 1 & 0 \\ \rho_{13} & 0 & 1 \end{bmatrix}$$

where $\mu$, $\sigma$, $\epsilon$ were chosen such that only ~5% of affinity-changing mutations are beneficial, as shown in experimental studies (**DeWitt et al., 2025**; **Kumar and Gromiha, 2006**; **Zhang and Shakhnovich, 2010**). $\Sigma$ is the correlation matrix and $\rho_{12}$ and $\rho_{13}$ parameterize the correlation of affinity changes between strains 1 and 2 and strains 1 and 3, respectively. The level of correlation is related to the level of conservation between the strains for B cell $j$'s target epitope, as described in the main text. We only consider correlation between the vaccine strain (strain 1) and historical strains (strains 2 and 3), but not between strains 2 and 3. Since B cells are selected for their affinity to strain 1, the correlation between strains 2 and 3 does not impact the nature of the antibody or memory B cell response.

In this study, we considered the correlation between three strains compared to two strains in our COVID model (**Yang et al., 2023**) since we examined the effect of sequential immunization on B cell responses against multiple unmatched influenza strains.

After the first immunization, pre-existing memory cells stochastically expand and differentiate. About 60% of newly proliferating memory cells become short-lived plasma cells that secrete antibodies (**Moran et al., 2018**). Memory cells can be activated whenever they encounter antigen, including at the T-B border, sub-capsular proliferative foci, medullar niches of the lymph node and spleen, and in the bone marrow (**Moran et al., 2018**; **Syeda et al., 2024**). Many types of antigen-presenting cells (APCs) can participate in this process. We coarse-grain these APCs into one cell type and call it 'FDCs'. Memory B cell expansion is also known to be T helper-cell dependent (**Syeda et al., 2024**). We refer to all locations outside GCs where memory cells are expanded in an antigen and T helper cell-dependent way as the EGC. In the EGC, the memory cells are selected for expansion in the same affinity-dependent way as GC B cells, except memory cells do not undergo mutation in our model. The number of helper T cells is set to its maximum value to reflect the faster kinetics of the EGC in the context of SARS-CoV-2 (**Goel et al., 2021**; **Moran et al., 2019**).

### Epitope masking

When epitope masking is considered, GC B cells specific for a particular epitope cannot capture antigen on FDCs if that epitope is bound to circulating antibodies. These circulating antibodies are produced by plasma cells from previous immunizations and expansion and differentiation of memory cells from previous immunizations. The amount of bound antigen is calculated using fast equilibrium of receptor–ligand binding:

$$[Ag_{masked}]^2 - \left( [Ag] + [Ig]^* + K_d^* \right) [Ag_{masked}] + [Ag] [Ig]^* = 0$$

where $[Ag_{masked}]$ is the concentration of masked (i.e. bound, antigen), $[Ig]^*$ is the total effective antibody concentration before masking, [Ag] is the total antigen concentration before masking, and $K_d^*$ is the effective average dissociation constant. The values in the equation above are calculated separately for each epitope. When we do not consider epitope overlap, $[Ig]^*$ and $K_d^*$ are exactly the concentration of antibodies targeting a particular epitope and the average dissociation constant of those antibodies. In the presence of epitope overlap, some antibodies can mask epitopes that spatially overlap with their primary target. In this case

$$\begin{bmatrix} [Ig_1]^* \\ [Ig_2]^* \\ [Ig_3]^* \end{bmatrix} = \begin{bmatrix} 1 & q_{12} & q_{13} \\ q_{12} & 1 & q_{23} \\ q_{13} & q_{23} & 1 \end{bmatrix} \begin{bmatrix} [Ig_1] \\ [Ig_2] \\ [Ig_3] \end{bmatrix},$$

where $[Ig_i]^*$ is the effective concentration of antibodies targeting epitope $i$ and $[Ig_1]$ is the actual concentration of antibodies targeting epitope $i$. $q_{mn}$ describes the overlap between epitope $m$ and $n$ and is the fraction of antibodies targeting epitope $m$ that can mask epitope $n$ (and vice versa). The effective average dissociation constant $K_{d,i}^*$ is calculated similarly:

$$\begin{bmatrix} [Ig_1]^*/K_{d,1}^* \\ [Ig_2]^*/K_{a,2}^* \\ [Ig_3]^*/K_{a,3}^* \end{bmatrix} = \begin{bmatrix} 1 & q_{12} & q_{13} \\ q_{12} & 1 & q_{23} \\ q_{13} & q_{23} & 1 \end{bmatrix} \begin{bmatrix} [Ig_1]/K_{d,1} \\ [Ig_2]/K_{d,2} \\ [Ig_3]/K_{d,3} \end{bmatrix}$$

The concentration of bound antigen $[Ag_{masked}]$ is then calculated and subtracted from the total antigen concentration since bound antigen cannot be seen by B cells. The resulting antigen concentration is then scaled such that the fraction of soluble antigen and fraction of antigen on the FDC match those fractions before epitope masking is not considered.

## Modifications to the original model

As a summary, the revised model includes three epitopes instead of two epitopes and considers three strains instead of two. This manifests in the distribution of naive B cells' germline affinities, the effective antigen concentration in epitope masking, and the correlated affinity changes in the GC. Using multiple epitopes and strains accounts for the diversity in influenza strains, each with distinct epitopes that may be conserved with the vaccine strain. By examining the relationships between multiple epitopes and strains, we gained general insights into factors that affect the number of immunizations needed to achieve a given level of coverage for historical and emergent strains, as detailed in the main text.

## Additional information

### Competing interests

Arup K Chakraborty: consultant for Flagship Pioneering, and it's affiliated company Aprioi Bio. He owns equity in these companies, and Metaphore Bio and Dewpoint Therapeutics. Daniel Lingwood: reports consulting activities for Metaphore Bio (a Flagship company), Tendel Therapies, Lattice Therapeutics Inc and BioMed X. The other authors declare that no competing interests exist.

### Funding

| Funder | Grant reference number | Author |
| --- | --- | --- |
| National Institutes of Health | AI155447 | Daniel Lingwood |
| National Institutes of Health | AI137057 | Daniel Lingwood |
| National Institutes of Health | AI153098 | Daniel Lingwood |

| Funder | Grant reference number | Author |
|---|---|---|
| National Institutes of Health | contract 75N93019C00052 | Daniel Lingwood |
| Massachusetts General Hospital | MGH Scholar Award | Daniel Lingwood |
| National Institutes of Health | AI060354 | Daniel Lingwood |
| National Institutes of Health | U19AI057229 | Arup K Chakraborty |
| The Ragon Institute of Mass General, MIT and Harvard | Schwartz AI Initiative | Yixiang Deng |
| Georgia Research Alliance | GRA Eminent Scholar | Ted M Ross |
| National Institutes of Health | AI146779 | Aaron G Schmidt |
| National Institutes of Health | AI089618 | Aaron G Schmidt |
| National Institutes of Health | contract 75N93019C00050 | Aaron G Schmidt |

The funders had no role in study design, data collection, and interpretation, or the decision to submit the work for publication.

## Author contributions

Yixiang Deng, Melbourne Tang, Formal analysis, Investigation, Visualization, Methodology, Writing – original draft, Writing – review and editing; Ted M Ross, Data curation, Writing – review and editing; Aaron G Schmidt, Visualization, Writing – review and editing; Arup K Chakraborty, Daniel Lingwood, Conceptualization, Resources, Supervision, Funding acquisition, Writing – original draft, Project administration, Writing – review and editing

## Author ORCIDs

Melbourne Tang ⬤ https://orcid.org/0009-0008-4341-7371
Arup K Chakraborty ⬤ https://orcid.org/0000-0003-1268-9602
Daniel Lingwood ⬤ https://orcid.org/0000-0001-5631-9238

## Ethics

This study uses de-identified data from a vaccine trial previously published by Nunez et al. (2017). The ethics approval statement for the vaccine trial is stated in that publication was as follows: The study procedures, informed consent, and data collection documents were reviewed and approved by was approved by the Western Institutional Review Board and the Institutional Review Boards of the University of Pittsburgh and the University of Georgia. Informed written consent was obtained from the parents/guardians of the children. The IRB number: WCG1292448. Notably, the study of Nunez et al. (2017) used commercial influenza vaccines and therefore does not fall under the Clinical Trials guidelines.

## Decision letter and Author response

Decision letter https://doi.org/10.7554/eLife.107042.sa1
Author response https://doi.org/10.7554/eLife.107042.sa2

## Additional files

### Supplementary files

MDAR checklist

Source data 1. Hemagglutination inhibition (HAI) values for the influenza virus strains, measured across longitudinal vaccine study (2013–2016) for n = 136 de-identified subjects >50 years of age and <38 years of age.

## Data availability

All longitudinal HAI values used in this study are provided in *Source data 1*. All original code and data files for the computational results have been deposited at https://github.com/mtang17/flu, (copy archived at *Tang, 2025*) and are publicly available.

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

## Appendix 1

**Appendix 1—table 1.** HA sequence information for the influenza stains in this study.

| Strain | Genbank | Type | Year | Location | GISAID |
|---|---|---|---|---|---|
| H1N1 A_Beijing_262_1995 | AB304819.1 | H1N1 | 1995 | Beijing | |
| H1N1 A_Brazil_11_1978 | HQ008267.1 | H1N1 | 1978 | Brazil | |
| H1N1 A_Brisbane_59_2007 | JN899402.1 | H1N1 | 2007 | Brisbane | |
| H3N2 A_Brisbane_10_2007 | KM978061.1 | H3N2 | 2007 | Brisbane | |
| B_Brisbane_60_2008 | FJ766842.1 | B | 2008 | Brisbane | |
| H1N1 A_California_07_2009 | NC_026433.1 | H1N1 | 2009 | California | |
| H1N1 A_California_10_1978 | CY021717.1 | H1N1 | 1978 | California | |
| H1N1 A_Chile_1_1983 | CY121261.1 | H1N1 | 1983 | Chile | |
| B_Colorado_06_2017 | CY236607.1 | B | 2017 | Colorado | |
| H1N1 A_Denver_1957 | CY146793.1 | H1N1 | 1957 | Denver | |
| B_Florida_4_2006 | EU515992.1 | B | 2006 | Florida | |
| H1N1 A_Fort_Monmouth_1_1947 | AF494250.1 | H1N1 | 1947 | Fort_Monmouth | |
| H3N2 A_Fujian_411_2002 | EU501153.1 | H3N2 | 2002 | Fujian | |
| B_Harbin_7_1994 | CY040441.1 | B | 1994 | Harbin | |
| B_Hong_Kong_330_2001 | AF532549.1 | B | 2001 | Hong_Kong | |
| H3N2 A_Hong_Kong_1_1968 | AF348177.1 | H3N2 | 1968 | Hong_Kong | |
| H3N2 A_Hong_Kong_4801_2014 | | H3N2 | 2014 | Hong_Kong | EPI1026711 |
| H3N2 A_Kentucky_UR07-0028_2008 | CY037791.1 | H3N2 | 2008 | Kentucky_UR07-0028 | |
| B_Lee_1940 | K00423.1 | B | 1940 | Lee | |
| B_Malaysia_2506_2004 | EU124275.1 | B | 2004 | Malaysia | |
| B_Massachusetts_2_2012 | MT056027.1 | B | 2012 | Massachusetts | |
| H1N1 A_Michigan_45_2015 | KY117023.1 | H1N1 | 2015 | Michigan | |
| H3N2 A_Mississippi_1_1985 | L19003.1 | H3N2 | 1985 | Mississippi | |
| H3N2 A_Nanchang_933_1995 | CY108293.1 | H3N2 | 1995 | Nanchang | |
| H1N1 A_New_Caledonia_29_1999 | DQ508857.1 | H1N1 | 1999 | New_Caledonia | |
| H1N1 A_New_Jersey_1976 | CY147422.1 | H1N1 | 1976 | New_Jersey | |
| H3N2 A_New_York_55_2004 | KM821338.1 | H3N2 | 2004 | New_York | |
| H3N2 A_Panama_2007_1999 | EF626612.1 | H3N2 | 1999 | Panama | |
| H3N2 A_Perth_16_2009 | GQ293081.1 | H3N2 | 2009 | Perth | |
| B_Phuket_3073_2013 | | B | 2013 | Phuket | EPI2195537 |
| H3N2 A_Port_Chalmers_12_1973 | CY113109.1 | H3N2 | 1973 | Port_Chalmers | |
| H1N1 A_Puerto_Rico_8_1934 | EF467821.1 | H1N1 | 1934 | Puerto_Rico | |
| H3N2 A_Shangdong_9_1993 | Z46417.1 | H3N2 | 1993 | Shangdong | |
| B_Sichuan_379_1999 | EF566113.1 | B | 1999 | Sichuan | |
| H3N2 A_Sichuan_30_1989 | CY108211.1 | H3N2 | 1989 | Sichuan | |
| H1N1 A_Singapore_6_1986 | D00406.1 | H1N1 | 1986 | Singapore | |
| H1N1 A_Solomon_Islands_03_2006 | EU100724.1 | H1N1 | 2006 | Solomon_Islands | |

*Appendix 1—table 1 Continued on next page*

*Appendix 1—table 1 Continued*

| Strain | Genbank | Type | Year | Location | GISAID |
|---|---|---|---|---|---|
| H1N1 A_South_Carolina_1_1918 | AF117241.1 | H1N1 | 1918 | South_Carolina | |
| H3N2 A_Switzerland_9715293_2013 | | H3N2 | 2013 | Switzerland | EPI814528 |
| H3N2 A_Sydney_5_1997 | KM821316.1 | H3N2 | 1997 | Sydney | |
| H1N1 A_Texas_36_1991 | DQ508889.1 | H1N1 | 1991 | Texas | |
| H3N2 A_Texas_1_1977 | EF626623.1 | H3N2 | 1977 | Texas | |
| H3N2 A_Texas_50_2012 | KC892952.1 | H3N2 | 2012 | Texas | |
| B_Texas_06_2011 | KC813979.1 | B | 2011 | Texas | |
| H1N1 A_USSR_90_1977 | HQ008265.1 | H1N1 | 1977 | USSR | |
| H3N2 A_Victoria_361_2011 | KM821347.1 | H3N2 | 2011 | Victoria | |
| H1N1 A_Weiss_1943 | CY147366.1 | H1N1 | 1943 | Weiss | |
| H3N2 A_Wisconsin_67_2005 | CY163704.1 | H3N2 | 2005 | Wisconsin | |
| B_Wisconsin_1_2010 | CY115183.1 | B | 2010 | Wisconsin | |
| B_Yamagata_16_1988 | M36105.1 | B | 1988 | Yamagata | |

**Appendix 1—table 2.** Simulation parameters.
Highlighted parameters were modified from the original model.

| Parameter | Value | Description | Note |
|---|---|---|---|
| **Antigen and antibody dynamics** | | | |
| $k_{Ig}$ | $0.8 \times 10^{-2}$ nM day$^{-1}$ PC$^{-1}$ | Rate of antibody production per plasma cell per day | Picked to match antibody titers at peak response to second SARS-CoV-2 vaccination (*Goel et al., 2021*; *Muecksch et al., 2022*) |
| $d_{Ig}$ | 0.025 day$^{-1}$ | Antibody decay rate | Picked to give antibody a half-life of ~28 days (*Goel et al., 2021*) |
| $d_{Ag}$ | 3 day$^{-1}$ | Antigen decay rate | Picked such that antigen decays rapidly in the first few days after vaccination (*Aung et al., 2023*; *Bhagchandani et al., 2024*; *Martin et al., 2021*; *Tam et al., 2016*) |
| $k_{deposit}$ | 1 hour$^{-1}$ | Rate of immune complex transport to FDC | Picked such that antigen is rapidly transported to the FDC within ~48 hours after vaccination (*Aung et al., 2023*; *Bhagchandani et al., 2024*; *Martin et al., 2021*) |
| $d_{Ic}$ | 0.15 day$^{-1}$ | Rate of decay of immune complex on FDC | Picked such that the secondary GCs last at least 3 months after SARS-CoV-2 vaccination as observed (*Gaebler et al., 2021*; *Muecksch et al., 2022*) |
| $[Ag]_0$ | 10 nM | | Picked within reasonable physiological ranges (*Martin et al., 2021*) |
| $[Ig]_0$ | 10$^{-2}$ nM | | Picked within reasonable physiological ranges (*Demonbreun et al., 2021*) |
| | 0 nM | Initial conditions | No initial IC exists |
| **B cell affinities** | | | |
| $N_{native}$ | 2000 cells/GC | Number of naïve B cells per GC | Based on the total number of naïve B cells (*Boyd and Joshi, 2014*; *Rees, 2020*) and frequency of naïve B cells that target the SARS-CoV-2 RBD (*Feldman et al., 2021*), divided across 200 GCs |
| $p_1$ | 0.8 | Fraction of naïve B cells that target epitope 1 | Tested for robustness in previous simulations (*Yang et al., 2023*). |
| $p_2$ | varied: 0.15, 0.18 | Fraction of naïve B cells that target epitope 2 | Varied in simulations |

*Appendix 1—table 2 Continued on next page*

*Appendix 1—table 2 Continued*

| Parameter | Value | Description | Note |
|---|---|---|---|
| $p_3$ | varied: 0.05, 0.02 | Fraction of naïve B cells that target epitope 3 | Varied in simulations |
| $E_1^h$ | 7 | Parameter for the germline affinity distribution of epitope 1. Affinity at which there is on average one naïve B cell targeting epitope 1 available for each GC | Tested for robustness in previous simulations (*Yang et al., 2023*) |
| $dE_{12}$ | 0.4 | Parameter for the germline affinity distribution of epitope 2. is the affinity at which there is on average one naïve B cell targeting epitope 2 available for each GC | Tested for robustness in previous simulations (*Yang et al., 2023*) |
| $dE_{13}$ | 0.8 | Parameter for the germline affinity distribution of epitope 3. is the affinity at which there is on average one naïve B cell targeting epitope 2 available for each GC | Picked such that the germline affinity distribution for epitope 3 has a shorter high-affinity tail than epitope 2, i.e. epitope 3 is less immunodominant than epitope 2 |
| $n_{res}$ | 80 | Length of string representation of B cell residues | Upper range of sum of CDR lengths in light and heavy chain (*Nowak et al., 2016*) |
| $\mu, \sigma, \epsilon$ | 3.1, 1.2, 3.08 | Parameters for shifted log-normal distribution that models the effect of affinity-changing mutations | Based on empirical distribution of affinity changes due to point-mutations in proteins (*Zhang and Shakhnovich, 2010*) |
| $\rho_{12}$ | epitope 1: 0.4 epitope 2: typically 0.95, varied: 0.6–0.95 epitope 3: 0.4 | Level of conservation between strain 1 and 2 | Picked such that a large portion of affinity-increasing mutations (~30 to 72%) in the conserved epitope (epitope 2) are beneficial for both strain 1 and 2;~19% for the other epitopes |
| $\rho_{13}$ | epitope 1: 0.4 epitope 2: 0.4 epitope 3: 0.95 | Level of conservation between strain 1 and 3 | Picked such that ~72% of affinity-increasing mutations in the conserved epitope (epitope 3) are beneficial for both strain 1 and 3;~19% for the other epitopes |
| $q_{12}$ | 0 without overlap 0.3 with overlap | Fraction of antibodies that target epitope 1 that can mask epitope 2 and vice versa (i.e. the amount of spatial overlap between epitope 1 and 2) | Varied in simulations; effects are also studied in previous simulations (*Yang et al., 2023*) |
| $q_{13}$ | 0 without overlap 0.3 with overlap | Fraction of antibodies that target epitope 1 that can mask epitope 3 and vice versa (i.e. the amount of spatial overlap between epitope 1 and 3) | Varied in simulations; effects are also studied in previous simulations (*Yang et al., 2023*) |
| $q_{23}$ | 0 | Fraction of antibodies that target epitope 2 that can mask epitope 3 and vice versa (i.e. the amount of spatial overlap between epitope 2 and 3) | Picked such that the subdominant epitopes (epitopes 2 and 3) are distinct |
| GC and EGC dynamics | | | |
| $C_0$ | nM | Reference antigen concentration | Tested for robustness in previous simulations (*Yang et al., 2023*) |
| $E_0$ | 6 | Reference binding affinity | Typical threshold for naïve B cell activation is (*Batista and Neuberger, 1998*) |

*Appendix 1—table 2 Continued on next page*

*Appendix 1—table 2 Continued*

| Parameter | Value | Description | Note |
|---|---|---|---|
| | typically 0.5 varied: 0.5, 0.7 | Stringency of selection of naïve and GC B cells by helper T cells based on amount of captured antigen | Varied in simulations; effects are also studied in previous simulations (*Yang et al., 2023*) |
| $N_{max}$ | 10 day$^{-1}$ | Approximately the maximum number of naïve B cells that can enter the GC per day | Based on experimental observation in mice for the number of GC B cells after 7 days (*Tas et al., 2016*) |
| $\beta_{max}$ | 2.5 day$^{-1}$ | Maximum rate of positive selection for GC and EGC B cells | Maximum proliferation is about ~4 times / day (*Tas et al., 2016*) |
| $\alpha$ | 0.5 day$^{-1}$ | Death rate of GC B cells | Picked to allow B cells to survive for ~2 days before death if they are not selected |
| $N_{T0}$ | 1200 | Maximum number of helper T cells involved in positive selection in GC and EGC | Picked such that GCs have a peak size of ~1,000 B cells for computational tractibility |
| $t_0$ | 14 days | Time at which number of helper T cells is maximal | Matches dynamics of T cell response to SARS-CoV-2 vaccination (*Goel et al., 2021*) |
| $d_T$ | 0.01 day$^{-1}$ | Death rate of helper T cells after time | Matches dynamics of T cell response to SARS-CoV-2 vaccination (*Goel et al., 2021*) |
| Plasma and memory cell dynamics | | | |
| $p_{exit}$ | 0.05 | Probability that a positively selected GC B cell exits and differentiates | Tested for robustness in previous simulations (*Yang et al., 2023*) |
| $p_{plasma}$ | 0.1 | Probability that a differentiating GC B cell becomes a plasma cell | Tested for robustness in previous simulations (*Yang et al., 2023*) |
| | 0.6 | Probability that a proliferating memory cell in EGC differentiates into a plasma cell | Based on observation in mice that ~60% of reactivated B memory cells differentiate into short-lived plasma cells (*Victora et al., 2010*) |
| $d_{PC}$ | 0.17 day$^{-1}$ | Death rate of plasma cells | Short-lived plasma cells have a half-life of ~4 days (*Moran et al., 2018*) |

