## [Editor Report]

How to develop the vaccination method to induce broadly reactive neutralizing antibodies against variant viruses such as influenza is now a central issue in this field. In this regard, this valuable study provides outlines the mechanism by which repeated vaccination broadens the breadth of antibody responses against epitope unmatched virus strains. The authors' mathematical model is solid and incorporates varipous parameteres that regulate B cell activation and antibody response.

---

## [Decision Letter]

**Decision letter after peer review:**

Thank you for submitting your article "Repeated vaccination with homologous influenza hemagglutinin broadens human antibody responses to unmatched flu viruses" for consideration by *eLife*. Your article has been reviewed by 2 peer reviewers, and the evaluation has been overseen by a Reviewing Editor and Aleksandra Walczak as the Senior Editor.

Essential Revisions:

Given the importance of GC reaction, incorporating the parameters for GC presitency would enhance the model to reflect the actual humoral responses.

*Reviewer #1 (Recommendations for the authors):*

Persistency of germinal center (GC) reaction in the secondary lymphoid organs plays a critical role in antibody diversification and affinity maturation, as exemplified in the case of SARS-CoV-2 mRNA vaccines, which induce robust and long-lasting GC responses in humans. Incorporating such parameters would enhance the model's ability to more accurately reflect the actual humoral immune responses observed in vaccinated and infected individuals.

---

## [Author Response]

Essential Revisions:Given the importance of GC reaction, incorporating the parameters for GC presitency would enhance the model to reflect the actual humoral responses.

We agree with the reviewer. Our simulations did account for this important phenomenon, but we did not explicitly note this. After booster shots, antigen-specific antibodies generated due to humoral processes from previous immunizations deposit more antigen on FDCs in GCs (Figure 5B). Recently we showed experimentally in a different context that enhanced antigen on FDCs at later time points is a consequence of past antibody responses [Bhagachandani et al., Science Immunology 2024]. This is an important reason that secondary GCs persist for onger times than GCs that form after primary immunization and our simulation results reflect this phenomenon. In our simulations, the GCs that form after primary immunization last for ~ 110 days, after the second immunization they persist for ~165 days, and then after subsequent immunizations last for ~ 180 days. We now provide an additional graph (Figure 5 —figure supplement 2; also reproduced below) in the revised manuscript, to explicitly note these results.

Reviewer #1 (Recommendations for the authors):Persistency of germinal center (GC) reaction in the secondary lymphoid organs plays a critical role in antibody diversification and affinity maturation, as exemplified in the case of SARS-CoV-2 mRNA vaccines, which induce robust and long-lasting GC responses in humans. Incorporating such parameters would enhance the model's ability to more accurately reflect the actual humoral immune responses observed in vaccinated and infected individuals.

We agree with the reviewer. Our simulations did account for this important phenomenon, but we did not explicitly note this. After booster shots, antigen-specific antibodies generated due to humoral processes from previous immunizations deposit more antigen on FDCs in GCs (Figure 5B). Recently we showed experimentally in a different context that enhanced antigen on FDCs at later time points is a consequence of past antibody responses [Bhagachandani et al., Science Immunology 2024]. This is an important reason that secondary GCs persist for onger times than GCs that form after primary immunization and our simulation results reflect this phenomenon. In our simulations, the GCs that form after primary immunization last for ~ 110 days, after the second immunization they persist for ~165 days, and then after subsequent immunizations last for ~ 180 days. We now provide an additional graph (Figure 5 —figure supplement 2; also reproduced below) in the revised manuscript, to explicitly note these results.